# Pathway to $\mathcal{O}(\sqrt{d})$ Complexity bound under Wasserstein metric of flow-based models

## Abstract

We develop general analytical tools to estimate the error of flow-based generative models under the Wasserstein metric and to establish an upper bound on the sampling iteration complexity, measured in terms of $\mathcal{O}(\sqrt{\operatorname{Tr} C})$, where $C$ is the covariance matrix of the prior distribution. When using the standard Gaussian as the prior, the posterior scales as $\mathcal{O}(\sqrt{d})$, which is shown to be optimal when the target distribution is Gaussian. More precisely, we show that the error under $W_2$ metric can be explicitly controlled by two parts: the Lipschitzness of the approximated push-forward maps of the backward flow, which scales independently of the dimension, and a local discretization error that scales with the square root of the trace of the covariance. The former one is related to the existence of Lipschitz changes of variables induced by the (heat) flow. The latter one consists of the regularity of the score function in both spatial and temporal directions.

Such framework is validated for the flow-based generative models associated with the Föllmer process and the 1-rectified flow. More precisely, we obtain the optimal $\mathcal{O}(\sqrt{d})$ complexity bound under the Gaussian-tail data assumption and, for learned models, Lipschitz-regular approximations of the score (or velocity field). Furthermore, we design numerical experiments to validate the optimality of the analysis with respect to discretization and spatial dimension.

## 1 Introduction

The landscape of deep learning has been fundamentally reshaped by the emergence of powerful generative models, including Generative Adversarial Networks (GANs) (Goodfellow et al., 2014; Arjovsky et al., 2017), Variational Auto-encoders (VAEs) (Kingma & Welling, 2014; Kingma et al., 2019), and Normalizing Flows (Papamakarios et al., 2021; Wang et al., 2023; Wan & Wei, 2022). These models have demonstrated remarkable success across diverse modalities such as images, audio, and text, by learning complex data distributions and generating high-quality samples (Achiam et al., 2023; Song et al., 2021).

Diffusion models (DM) are the state-of-the-art generative models, which can be analyzed via the SDE framework (Song et al., 2021). With the same forward and backward marginal as DM, flow-based models (Chen et al., 2023b;c) are generative models with deterministic flow given initial distribution, offering a principled framework for generative modeling and statistical inference (Cheng et al., 2024).

From a theoretical perspective, convergence guarantees of generative models are typically established under distributional discrepancies such as KL divergence, total variation, or Wasserstein distance. Early works primarily focus on KL-based analysis (Chen et al., 2023a; Benton et al., 2024; Conforti et al., 2025a; Li et al., 2024), which enables tractable error control via tools such as Girsanov's theorem. However, in many practical applications involving structured data (e.g., images or biological data), the target distribution often lies on a low-dimensional or compact sub-manifold of the ambient space (Tenenbaum et al., 2000; Bengio et al., 2017). In such settings, KL divergence becomes ill-defined or fails to capture meaningful geometric discrepancies.

This motivates the study of generative models under the Wasserstein metric, particularly the $W_2$ distance, which remains well-defined under finite second moments and is better suited for structured distributions. However, analyzing convergence under $W_2$ is significantly more challenging. Unlike KL-based methods, Wasserstein analysis suffers from the accumulation of local discretization errors in Lyapunov-type estimates due to the lack of contractivity in reverse-time dynamics. Existing works mitigate this issue by imposing strong structural assumptions such as log-concavity (Gao & Zhu, 2025; Bruno & Sabanis, 2025). In particular, Proposition 6 in Gao et al. (2025) establishes that for the standard Gaussian as target distribution, $\mathcal{O}(\sqrt{d})$ complexity bound is optimal. In efforts to obtain complexity bounds under assumptions more general than log-concavity, recent works Bruno & Sabanis (2025) derived an $\mathcal{O}(d)$ bound using the weakly log-concave assumption Conforti (2024); Conforti et al. (2025b), and Gentiloni-Silveri & Ocello (2025) obtained an $\mathcal{O}(d^2)$ bound under the similar assumption. Despite Chen et al. (2023b) claiming the ODE type flow is provably fast by showing an $\mathcal{O}(\sqrt{d})$ bound under a predictor-corrector scheme, the analytical pathway to $\mathcal{O}(\sqrt{d})$ complexity upper bound for practical ODE flow based models in the non-log-concave setting remains unclear.

In light of this, **the main contribution of this paper** is to provide analytical tools that study the accumulation error along the sampling flow under the Wasserstein metric and hence ensure the optimal iteration complexity bound $\mathcal{O}(\sqrt{d})$. Our analysis reveals that the $W_2$ distance between the generated and target distributions is governed by the Lipschitz regularity of the push-forward maps induced by the sampling flow. In addition, we identify the temporal regularity of the velocity field, namely $\partial_t V$, as a key factor controlling the local truncation error and thus the overall complexity. To establish these results, we introduce a Gaussian-tail assumption that ensures well-posedness and dimension-free Lipschitz bounds for the Föllmer flow. Under this condition, we obtain explicit control of both spatial and temporal regularity, leading to an optimal iteration complexity of $\mathcal{O}(\sqrt{d})$ without requiring log-concavity; summarized as below.

**Theorem 1** (informal, see Theorem 11 for the formal version.). *Under Gaussian-tail assumption and sufficiently accurate Lipschitz velocity approximations, we show that*

$$W_2(P_{\text{data}}, Q) = \mathcal{O}(\sqrt{d}\,h + \varepsilon),$$

*where $P_{\text{data}}$ is the target distribution, $Q$ is the induced distribution, $d$ is the dimension, and $h$ is the discretization step size.*

Table 1: Complexity bounds for DM/flow-based models in $d$ dimensions: previous results vs. ours. The complexity bound is measured by number of steps needed to ensure error under Wasserstein metric is less than $\epsilon_0$.

| Target $P_0$ | Complexity | Result |
|---|---|---|
| $P_0$ log-concave* | $\mathcal{O}\left(\frac{\sqrt{d}}{\epsilon_0}(\log \frac{d}{\epsilon_0})^2\right)$ | Gao & Zhu (2025) Tab. 1 |
| G-tail Ass.* | $\mathcal{O}\left(\frac{\sqrt{d}}{\epsilon_0}\log \frac{d}{\epsilon_0^2}\right)$ | Wang & Wang (2024) Cor. 3.5 |
| one-side Lip+weakly log-concave* | $\mathcal{O}\left(\frac{d^2}{\epsilon_0^2}\right)$ | Gentiloni-Silveri & Ocello (2025) Thm. 3.5 |
| weakly log-concave* | $\mathcal{O}\left(\frac{d}{\epsilon_0^2}\right)$ | Bruno & Sabanis (2025) Thm. 3.12 |
| **G-tail** Ass. 7 | $\mathcal{O}\left(\frac{\sqrt{d}}{\epsilon_0}\right)$ | Our Thm. 11 |

* denotes works on diffusion models.

## 1.1 Related Work

Lipschitz transport and flow regularity. The study of Lipschitz transport maps originates from Caffarelli's contraction theorem (Caffarelli, 2000) and has been extended to various settings including compact support (Colombo et al., 2017), diffusion processes (Kim & Milman, 2012), and Föllmer flows (Dai et al., 2023; Mikulincer & Shenfeld, 2023). Recent works further improve Lipschitz bounds under weaker assumptions (Neeman, 2022; Fathi et al., 2024; Brigati & Pedrotti, 2024). For clarity, we summarize the assumptions on target distributions and their Lipschitz constants in Table 2 of Appendix A. Despite these results shed light on potential minimal assumption for the convergence guarantee of flow-based models, in later context,

we will demonstrate that estimation of the time derivative of velocity field $\partial_t V$ is also crucial on the pathway of optimal complexity bounds.

Continuous generative flows. Building on score-based (Song et al., 2021) and diffusion models (Gao & Zhu, 2025), recent works improve sampling efficiency via distillation (Salimans & Ho, 2022) and continuous flow formulations. Flow matching (FM) (Lipman et al., 2023) extends continuous normalizing flows (CNFs) (Chen et al., 2018) by learning neural ODE velocity fields that match prescribed probability paths, unifying diffusion and non-diffusion models. Rectified flow (Liu et al., 2023; Rout et al., 2024) further constructs nearly straight transport trajectories, enabling efficient one-step generation. Stochastic interpolants (Albergo et al., 2023; Albergo & Vanden-Eijnden, 2023) provide a unified framework bridging flow-based and diffusion models, recovering the Schrödinger bridge when optimized (Léonard, 2013). More recently, Flow Map Matching (FMM) (Boffi et al., 2025) accelerates sampling via two-time flow maps, while Geng et al. (2025) connects one-step generative modeling with multiscale physical simulations, achieving strong performance on ImageNet $256 \times 256$.

Convergence bounds. Recent studies establish convergence guarantees under KL, $W_2$, or TV distances. For example, Albergo & Vanden-Eijnden (2023) bounds the $W_2$ error under Lipschitz assumptions, while Gao et al. (2024) and Cheng et al. (2024) provide KL-based complexity results. While these works deriving the Wasserstein error bounds with the assumption of Lipschitz-ness, the assumption in Proposition 6 of this work is more realistic and consistent with the estimates for the continuous flow.

## 1.2 Contributions

- We point out that the $W_2$-distance between the generative and target distributions is controlled by the Lipschitzness of the push-forward maps introduced by sampling flow. By providing concrete bounds on the Lipschitz coefficient, we obtain an explicit estimate of the accumulation error.

- While prior works often rely on smoothness or strict log-concavity, we adopt a general condition in applications-the Gaussian-tail Assumption 7 to provide the well-posedness and Lipschitz regularity of Föllmer flow, with explicit, dimension-free Lipschitz bounds (Corollary 23 and Corollary 24).

- By leveraging the Gaussian-tail Assumption 7 to obtain accurate upper bounds on the time derivative of velocity field $|\partial_t V|$ (Theorem 8), our framework avoids the need for end-point constraints or early stopping (Lyu et al., 2022), enabling training and sampling throughout the entire interval $t \in [0, 1]$. This framework naturally extends the $\mathcal{O}\left(\sqrt{d}\right)$ complexity results of the SDE flow to the deterministic flow, achieving even better complexity than previous approaches (Wang & Wang, 2024).

- Using an Föllmer flow sampler with an explicit score function, we study the Euler discretization error and observe first-order convergence in time. The experiments further confirm the predicted dimension scaling $N(d) \sim d^{0.5}$, with a fitted slope of approximately 0.472 in log–log regression.

## 2 Flow-based Model

We begin by introducing a unified formulation of flow-based generative models. This general framework allows the convergence analysis in Section 3 to apply seamlessly to both the Föllmer flow and more general sampling dynamics. Consider a continuous flow governed by a velocity field $V$ via the ODE[1]

$$\frac{\mathrm{d}\overleftarrow{X}_t}{\mathrm{d}t} = V\left(t, \overleftarrow{X}_t\right), \quad \overleftarrow{X}_0 = x, \quad t \in [0, 1]. \tag{1}$$

With the $N$ steps discretization in time, $0 = t_0 < t_1 < \cdots < t_N = 1$, the ODE equation 1 in each sub-interval $[t_n, t_{n+1}]$, can be interpreted as a local transport map,

$$T_n(\overleftarrow{X}_{t_n}) = \overleftarrow{X}_{t_{n+1}}. \tag{2}$$

---

[1]We used the left arrow $\overleftarrow{\cdot}$ to represent its connections to the backward process in the score-based model.

The overall flow-based model $\overleftarrow{X}_1(x)$ is then obtained by the composition of transport maps

$$\overleftarrow{X}_1(x) = (T_{N-1} \circ T_{N-2} \circ \cdots \circ T_0)(x).$$

An approximation of $\overleftarrow{X}_1(x)$ can be interpreted as approximation of $\{T_n\}_{n=0}^{N-1}$ by $\{\widetilde{T}_n\}_{n=0}^{N-1}$. To quantify the error of the approximation, we denote the marginal distribution of the actual state $\overleftarrow{X}_{t_{n+1}}$ by $\overleftarrow{P}_{t_{n+1}}$, and $\overleftarrow{Q}_{t_{n+1}}$ of the approximated state. Correspondingly we have,

$$\overleftarrow{P}_{t_{n+1}} = (T_n)_\#(\overleftarrow{P}_{t_n}), \quad \overleftarrow{Q}_{t_{n+1}} = (\widetilde{T}_n)_\#(\overleftarrow{Q}_{t_n}). \tag{3}$$

In this work, we mainly take the Föllmer flow (Dai et al., 2023) as an concrete example under the assumptions over target distribution $P_{\text{data}}$; see Appendix B.10 for an application to 1-rectified flow (Liu et al., 2023).

**Föllmer flow**  For any $\varepsilon \in (0,1)$, we consider a diffusion process $(\overrightarrow{X}_t)_{t\in[0,1-\varepsilon]}$ that gradually transforms the target distribution $\nu$ into a centered Gaussian distribution $\mathcal{N}(0,C)$ with covariance matrix $C$ over time by the following Itô SDE

$$d\overrightarrow{X}_t = -\frac{1}{1-t}\overrightarrow{X}_t\,dt + \sqrt{\frac{2C}{1-t}}\,dW_t, \quad \overrightarrow{X}_0 \sim \nu, \quad t \in [0, 1-\varepsilon], \tag{4}$$

where $W_t$ is a standard Brownian motion, $C$ is a symmetric, positive-definite covariance matrix. We allow a general covariance matrix $C$ to cover the standard isotropic Gaussian setting $C = I_d$ commonly used in computer vision applications, while also accommodating anisotropic Gaussian reference measures with $C \neq I_d$. This generality enables the treatment of correlated Gaussian structures and facilitates extensions to infinite-dimensional formulations requiring compactification (Lim et al., 2025). The transition probability distribution from $\overrightarrow{X}_0$ to $\overrightarrow{X}_t$ is given by

$$\overrightarrow{X}_t | \overrightarrow{X}_0 = x_0 \sim \mathcal{N}\big((1-t)x_0, t(2-t)C\big). \tag{5}$$

The probability distribution function $(\bar{p}_t)_{t\in[0,1-\varepsilon]}$ of the forward diffusion process satisfies the Fokker-Planck-Kolmogorov (FPK) equation in an Eulerian framework

$$\partial_t \bar{p}_t = \nabla \cdot \left( \bar{p}_t \cdot \frac{1}{1-t}[x + C\nabla \log \bar{p}_t(x)] \right) \quad \text{on } [0, 1-\varepsilon] \times \mathbb{R}^d, \quad \bar{p}_0 = \nu. \tag{6}$$

Then Föllmer flow is formally defined as the backward process of such a forward diffusion equation 4, while preserving the same marginal distributions in equation 6.

**Definition 2** (Föllmer flow in formal sense). *A Föllmer flow $(\overleftarrow{X}_t)_{t\in[0,1]}$ solves the IVP*

$$\begin{cases} \frac{d\overleftarrow{X}_t}{dt} = V(t, \overleftarrow{X}_t), \quad \overleftarrow{X}_0 \sim \gamma_C, \quad t \in [0,1], \\ V(t,x) := \frac{1}{t}[x + S(t,x)], \quad \forall t \in (0,1]; \qquad V(0,x) := \sqrt{C}\mathbb{E}_\nu[X], \end{cases} \tag{7}$$

*$S(t,x) := C\nabla \log p_t(x)$ is the score function with probability density $p_t = \bar{p}_{1-t}$ in forward FKP equation equation 6. We call $V(t,x)$ a Föllmer velocity field.*

Following equation 3, we define $\overrightarrow{P}_{t_n}$ as the marginal distribution of $\overrightarrow{X}_{t_n}$ in the forward diffusion process. Given the initial distribution $\overrightarrow{P}_0 = P_{\text{data}}$, then for all $t \in [0,1]$, $\overleftarrow{P}_{t_n} = \overrightarrow{P}_{1-t_n}$.

In practice, the velocity field $V(1-t,x) = \frac{1}{1-t}[x + C\nabla \log \bar{p}_t(x)]$ is not available since no closed form expression of $\bar{p}_t$ is known. To this end, one approximates $V$ by a neural network $\widetilde{V}$. The network is trained by minimizing an $\mathbb{L}_2$ estimation loss,

$$\mathbb{E}_{\bar{p}_t(x)} \left\| \widetilde{V}(1-t, x) - \frac{1}{1-t}[x + C\nabla \log \bar{p}_t(x)] \right\|^2. \tag{8}$$

For simplicity, we introduce the notation $X_t := (1 - t)X_0 + \sqrt{t(2 - t)C}\,\mathcal{N}$, which shares the same marginal distribution as $\overrightarrow{X}_t$ in equation 5. Then the velocity field $V(1 - t, x)$ can be expressed as a conditional expectation (Yubin et al., 2025),

$$V(1 - t, X) := \frac{1}{1 - t}\big[X + S(1 - t, X)\big] = \mathbb{E}_{X_0|X_t}\left[\frac{1}{1 - t}X_t - \frac{X_t - (1 - t)X_0}{(1 - t)t(2 - t)}\,\middle|\, X_t = X\right].$$

With an appropriate weight of the $t$-variable, the loss in equation 8 becomes an approximation of this conditional expectation via mean-squared prediction error,

$$\mathbb{E}_{X_0,\, N\sim\mathcal{N}(0,I_d),\, t}\left[\lambda(t)\left\|\widetilde{V}(1 - t, X_t) - \frac{1}{1 - t}X_t + \frac{\sqrt{C}\mathcal{N}}{(1 - t)\sqrt{t(2 - t)}}\right\|^2\right]. \tag{9}$$

After training, with $\widetilde{V}(1 - t, x)$, one can generate samples of the target distribution via an Euler-type discretization of the continuous-time process, starting from the Gaussian initialization $\gamma_C := \mathcal{N}(0, C)$,

$$\frac{\mathrm{d}\overleftarrow{Y}_t}{\mathrm{d}t} = \widetilde{V}(t_n, \overleftarrow{Y}_{t_n}), \quad \overleftarrow{Y}_{t_0} \sim \gamma_C, \quad t \in [t_n, t_{n+1}], \quad n = 0, 1\ldots, N - 1. \tag{10}$$

Note that equation 10 defines the transport map $\widetilde{T}_n$ for the learned Föllmer flow, governed by the approximate velocity field $\widetilde{V}(t_n, \overleftarrow{Y}_{t_n})$ over the sub-interval $[t_n, t_{n+1}] \subset [0, 1]$. Distribution of generation $\overleftarrow{Q}_t$ is then defined by equation 3.

**Well-posedness of Föllmer flow**   Under appropriate assumptions on the target distribution $\nu$, one can show the Föllmer flow being the time-reversal of the forward diffusion process equation 4. In the isotropic setting ($C = I_d$), Dai et al. (2023) introduced the score function

$$S(t, x) := \nabla \log \int_{\mathbb{R}^d} (2\pi(1 - t^2))^{-\frac{d}{2}} \exp\left(-\frac{|x - ty|^2}{2(1 - t^2)}\right) \nu\,(\mathrm{d}y).$$

It can be shown that the velocity field $V$ is Lipschitz continuous in $x$ with a well-defined initial condition $V(0, x)$. By the Cauchy-Lipschitz theory (Ambrosio & Crippa, 2014), one can define a Lagrangian flow $(X_t^*)_{t\in[0,1]}$ governed by the well-posed ODE system,

$$\mathrm{d}X_t^* = -V(1 - t, X_t^*)\mathrm{d}t, \quad X_0^* \sim \nu, \quad t \in [0, 1],$$

sharing the same marginal distribution with equation 4.

For the regularity and well-posedness analysis, Dai et al. (2023) established the result under the third moment assumption, semi-log-convexity, and additional structural conditions on $\nu$ in the isotropic setting ($C = I_d$). In this work, we only impose the Gaussian-tail Assumption 7, which automatically guarantees finite moments of all orders, including the third-moment condition used throughout the analysis (taking $p = 3$ in equation 17 of Appendix B.2). This eliminates the need for additional moment assumptions and allows for a spatially anisotropic Gaussian initialization ($C \neq I_d$). Under these assumptions, we establish the regularity of the velocity field (Theorem 8) and prove the well-posedness of the corresponding Föllmer flow (Lemma 19 in Appendix B.3).

**General Notations**   Let $\gamma_C$ denote the density of $\mathcal{N}(0, C)$. For an $n \times n$ matrix $A$, the operator norm $\|\cdot\|$ is defined as

$$\|A\| = \sup_{v\neq 0} \frac{|Av|}{|v|} = \text{largest eigenvalue of } \sqrt{A^T A}.$$

For symmetric positive-definite $A$, define the weighted $\ell_2$ norm

$$|x|_A^2 := (A^{-1/2}x, A^{-1/2}x),$$

which reduces to the standard $\ell_2$ norm $|\cdot|$ when $A = I$. For a vector (matrix)-valued function $f(x)$,

$$|f|_\infty = \sup_x |f(x)|, \quad (\|f\|_\infty = \sup_x \|f(x)\|).$$

# 3 General Theory to the optimal complexity bound

In this section, we present the main results. Our analysis begins with a general flow-based framework, through which we develop Wasserstein-based analytical tools that yield an optimal iteration complexity bound of $\sqrt{d}$. We then validate the assumptions and present the complexity results for the Föllmer flow and 1-rectified flow under the Gaussian-tail assumption.

## 3.1 Lipschitz changes of variables imply Wasserstein bound of flow-based models

For the sake of compactness, we impose the following assumption on the second-order moment.

**Assumption 3** (Second moment). *The data distribution has a bounded second moment, $M_2 := \mathbb{E}_{p_0}|x|^2 < \infty$. In addition, denote*

$$M_0 = \max\{\mathrm{Tr}(C), M_2\},$$

*relates to the maximum second-order moment, and recall $C$ is a symmetric, positive-definite covariance matrix for the initial Gaussian distribution of the Föllmer flow equation 4.*

In the isotropic case $C = I_d$, by $\mathrm{Tr}(C) = d$ and thus $M_0 = \mathcal{O}(d)$, we aim to obtain the optimal dependence. At the same time, we retain $C \neq I_d$ in the derivation to extend our theory to infinite-dimensional settings (Lim et al., 2025), with the general case further discussed in Appendix B.11 for Bayesian inverse problems.

Next, we make three assumptions, each holding with some dimension-free constants. We regard these assumptions as generally valid, and under them, our convergence result Proposition 6 can be established.

**Assumption 4** (Lipschitzness of transport maps). *The exact and approximate transport maps satisfy the following Lipschitz regularity conditions.*
*(i) For every $n = 0, \ldots, N - 1$, the map $T_n$ is Lipschitz continuous with $\mathrm{Lip}(T_n) < \infty$, and the cumulative Lipschitz constant satisfies $\prod_{j=0}^{n} \mathrm{Lip}(T_j) < \infty$.*
*(ii) For every $n = 0, \ldots, N - 1$, the approximated map $\widetilde{T}_n$ of $T_n$ is Lipschitz continuous with $\mathrm{Lip}(\widetilde{T}_n) < \infty$, and the cumulative Lipschitz constant satisfies $\prod_{j=0}^{n} \mathrm{Lip}(\widetilde{T}_j) < \infty$.*

Assumption 4 (i) will be verified in Corollary 23 using the Lipschitz regularity of the exact velocity field established in Theorem 8, while Assumption 4 (ii) will be verified in Corollary 24 based on the Lipschitz regularity of the learned velocity field imposed in Assumption 10. The final assumption concerns the local discretization error between $T$ and $\widetilde{T}$ at each time step $h$, as described below.

**Assumption 5** (Accuracy of approximation). *There exists constants $\overline{K}, \overline{K_1}, \overline{K_2}, \epsilon$, such that*

$$\sqrt{\mathbb{E}_{x \sim \overleftarrow{P}_{t_n}} |\widetilde{T}_n(x) - T_n(x)|^2} \leq h\left( \left( \overline{K}\sqrt{M_0} + \frac{\overline{K_1}}{\sqrt{1 - t_n^2}} + \overline{K_2}\right)h + \epsilon \right),$$

*with time step size $h = t_{n+1} - t_n$.*

This scaling follows since

$$\widetilde{T}_n(x) - T_n(x) = h\big(V(x) - \widetilde{V}(x)\big) + \mathcal{O}(h^2),$$

as verified in the Föllmer case Theorem 11. The term $\mathcal{O}(h)$ reflects the $\epsilon$-accuracy of the learned velocity $\widetilde{V}(x)$ (Assumption 9), while the term $\mathcal{O}(h^2)$ stems from the Taylor expansion of $T_n(x)$ over $[t_n, t_{n+1}]$ and depends on its regularity, possibly also on ambient dimension $d$ and time $t$.

Next, we outline the core proof strategy of this work. The key step is to demonstrate the Lipschitz continuity of both the original and discretized flows, which is critical for guaranteeing the convergence of flow-based generative models.

**Proposition 6.** *Assume that the target distribution satisfies Assumption 3 and follows Lipschitzness Assumption 4 and approximation error Assumption 5. With constant step size $h$, the Wasserstein-2 distance*

*between the target distribution $\overrightarrow{P}_0 = \overleftarrow{P}_1$ and the generation $\overleftarrow{Q}_1$ is bounded as,*

$$
\begin{aligned}
\mathcal{W}_2(\overleftarrow{P}_1, \overleftarrow{Q}_1) \leq & \left( \prod_{j=0}^{N-1} \mathrm{Lip}(\widetilde{T}_j) \right) \mathcal{W}_2(\overleftarrow{P}_0, \overleftarrow{Q}_0) \\
& + h \sum_{k=0}^{N-2} \prod_{j=k}^{N-2} \mathrm{Lip}(\widetilde{T}_j) \left( \left( \overline{K}\sqrt{M_0} + \frac{\overline{K_1}}{\sqrt{1-t_j^2}} + \overline{K_2} \right) h + \epsilon \right).
\end{aligned}
\tag{11}
$$

Proof see Appendix B.1.

Proposition 6 provides a general error-propagation framework for Wasserstein dynamics under Lipschitz transport maps. It shows that the first term in the bound scales the initial discrepancy $\mathcal{W}_2(\overleftarrow{P}_0, \overleftarrow{Q}_0)$ by the product of Lipschitz constants $\prod_{j=0}^{N-1} \mathrm{Lip}(\widetilde{T}_j)$, and the second term $\left( \overline{K}\sqrt{M_0} + \frac{\overline{K_1}}{\sqrt{1-t_j^2}} + \overline{K_2} \right) h + \epsilon$, captures accumulated discretization errors (Assumption 5) and a local discretization error scales $\mathcal{O}(\sqrt{M_0})$, yielding the $\mathcal{O}(\sqrt{d})$ dependence in the isotropic case $C = I_d$. To be noted, in the limit of $h \to 0$, $h \sum_{k=0}^{N-2} \frac{1}{\sqrt{1-t_k^2}} \to \frac{\pi}{2}$. Similar results are listed in (Albergo & Vanden-Eijnden, 2023; Boffi et al., 2025), while the precise scaling of the second term remains unspecified.

Building on this framework, our main contribution is its specialization to the Fölmer flow under the Gaussian-tail Assumption 7, and deriving dimension-explicit regularity estimates.

### 3.2 Analyses of Föllmer flow under Gaussian-tail assumption

In this section, we specialize the general convergence framework established in Proposition 6 to the *Föllmer flow* and derive the corresponding convergence result via Lipschitz changes of variables. We stress that Proposition 6 itself applies to a broader class of continuous-flow matching models and is not specific to the Föllmer construction; see Appendix B.10 for an application to 1-rectified flow.

Our analysis is based on the following key assumption that the tail distribution of the target is similar to a Gaussian distribution with covariance matrix $A$.

**Assumption 7** (**Gaussian-tail**). *The probability density function of target distribution $\bar{p}_0 \in C^2(\mathbb{R}^d)$ and admits the following tail decomposition:*

$$
\bar{p}_0(x) = \exp\left( -\frac{|x|_A^2}{2} \right) \exp(h(x)),
\tag{12}
$$

*where there are independent of dimension constants such that,*

*(i) A is a symmetric, positive-definite matrix which can be simultaneously diagonalized with C, and*

$$
\|A\| < \infty, \quad \|C\| < \infty, \quad \|AC^{-1}\| < \infty, \quad \|CA^{-1}\| < \infty.
$$

*(ii) the remainder term h follows*

$$
|\sqrt{C}\nabla h|_\infty < \infty, \quad \|C\nabla^2 h\|_\infty < \infty.
$$

The Gaussian-tail Assumption 7 generalizes the log-concavity condition in (Ding et al., 2023; Gao et al., 2024) to heavier-than-sub-Gaussian-tails while ensuring sufficient decay for well-posedness and convergence. Compared to sub-Gaussian tail assumptions (Wainwright, 2019) as well as the weak semi-log-concavity assumption of (Chaintron et al., 2025; Bruno & Sabanis, 2025), it imposes stronger control on the tail behavior. Nevertheless, this stronger regularity enables sharper complexity guarantees: the latter one implies $\mathcal{O}(d)$ sampling complexity, whereas the Gaussian-tail assumption achieves $\mathcal{O}(\sqrt{d})$. We note that the dimension-free

constants in the velocity field regularity (Theorem 8) and, consequently, in the convergence bound (Theorem 11) are not intrinsic to arbitrary target distributions; rather, they depend on the compatibility between the Gaussian-tail matrix $A$ and the covariance matrix $C$ of the Gaussian reference measure as imposed in Gaussian-tail Assumption 7. Such consistency assumptions are also common in the high-dimensional MCMC and Bayesian inverse-problem literature, where dimension-robust algorithms typically require the proposal or reference Gaussian measure to be compatible with the target distribution; see, for example, the preconditioned Crank–Nicolson algorithm in Cotter et al. (2013). Notably, the Gaussian-tail Assumption 7 admits a non-log-concave setting and also accommodates realistic distributions such as early stopping and extendable to infinite-dimension; see Theorem 11, Theorem 15, and Theorem 26.

Similar Gaussian-tail-type decompositions of $\bar{p}_0$ in equation 12 also appeared in (Cole & Lu, 2024; Pidstrigach et al., 2024), primarily for establishing well-posedness and generalization of the sampling process. In contrast, our Assumption 7 is further tailored to analyze convergence of the discrete-time dynamics and its dimensional dependence; while slightly more restrictive, it ensures that the following theorem provides bounds on the Lipschitz constant and the time derivative of the Föllmer velocity field in equation 7. These bounds, in turn, support the Lipschitz change-of-variables argument in Corollary 23 and the convergence rate established in Theorem 11.

**Theorem 8** (Regularity of the velocity field). *The Gaussian-tail Assumption 7 implies the Föllmer velocity field $V(t, \cdot)$ has the following regularity properties:* $\forall t \in [0, 1]$,

$$
\begin{aligned}
|V(t, x)| &\leq K_0 + K_2 t |x|, \\
\|\nabla V(t, \cdot)\|_\infty &\leq (K_1 + K_2) t, \\
|\partial_t V(t, x)| &\leq K_5 |x| + \frac{K_6}{\sqrt{1 - t^2}} + K_7,
\end{aligned}
\tag{13}
$$

*where the coefficients are dimension-free constants, given explicitly in Table 4 of Appendix C.*

The quantity $|\partial_t V(1-t, x)|$, which characterizes the temporal variation of the score along the forward process, is commonly referred to as score perturbation bound (Chen et al., 2023b). Controlling this term is essential for convergence, as its local behaviour in the spatial and temporal domains may lead to instability. Existing works, therefore, combine time-domain restriction with early stopping to ensure sufficient regularity after a short amount of forward diffusion. For instance, to handle the blow-up of $|\partial_t V(t, x)|$ near $t = 0, 1$, Ding et al. (2023) restrict $t$ to $[\delta, 1 - \delta]$. In particular, Gao et al. (2024) shows Lipschitz continuity of $V$ in $t$ over $[0, 1 - \delta_0]$ with constant scaling as $\mathcal{O}(\delta_0^{-2})$. In contrast, under our Gaussian-tail assumption, the control over the second derivative of the tail allows us to bound $|\partial_t V(t, x)|$ using techniques such as the Brascamp-Lieb inequality (Brascamp & Lieb, 1976). Such analysis reveals that the temporal term $\frac{1}{\sqrt{1-t^2}}$ is integrable on $[0, 1]$ and the spatial term is $\mathcal{O}(|x|)$. To our knowledge, it is the first to yield the improved $\mathcal{O}(\sqrt{d})$ complexity bound for the probability flow ODE in practice (without corrector terms if compared with (Chen et al., 2023b)); c.f. Corollary 12. Notably, this bound is dimensionless even in the presence of an anisotropic assumption ($C \neq I_d$). This follows from the fact that all constants entering the estimate are fully explicit and depend only on the Gaussian-tail regularity quantities in Assumption 7, including operator-norm bounds associated with $A$ and $C$ and the quantities $|\sqrt{C}\nabla h|_\infty$ and $\|C\nabla^2 h\|_\infty$; see Table 4. In particular, no additional dependence on the ambient dimension is hidden in these constants. This highlighting a qualitative improvement over prior analyses that rely on isotropic regularization or compact-support assumptions. Detailed proof of Theorem 8 is provided in Appendix B.4. Moreover, Appendix B.5 shows that the same estimate extends to an averaged Föllmer velocity inspired by MeanFlow (Geng et al., 2025), preserving linear growth under temporal averaging.

### 3.3 Main Convergence theories

Based on the regularity of the Föllmer velocity field established in Theorem 8, we impose Assumptions 9 and 10 to control the approximation error at the discretization points and to ensure that the learned velocity field inherits the regularity of the continuous flow under the Gaussian-tail Assumption 7.

**Assumption 9** (Accuracy of the learned velocity field). *For each time discretization point $t_n$, the accuracy of learned velocity $\widetilde{V}(t_n, x)$ approximates the true velocity field $V(t_n, x)$ with uniformly bounded error in*

*expectation:*

$$\mathbb{E}_{\overrightarrow{P}_{1-t_n}} |V(t_n, x) - \widetilde{V}(t_n, x)|^2 \leq \epsilon^2.$$

**Assumption 10** (Regularity of the learned velocity field)**.** *Assume the learned velocity field $\widetilde{V}(t, x)$ follows*

$$\|\nabla \widetilde{V}(t_n, \cdot)\|_\infty \leq (K_1 + K_2 + K_8)t_n$$

*for some positive constant $K_8$.*

Regularity assumptions on learned vector fields are common in convergence analyses of score-based generative models. For example, Bruno & Sabanis (2025); Bruno et al. (2023) assume global Lipschitz continuity of the learned score function with respect to the input variable together with Lipschitz continuity in time (Assumption 3.a), while Huang et al. (2025) further requires bounded first- and second-order derivatives of the learned score function (Assumption 3.3) to control discretization errors for probability flow ODEs. Assumption 10 serves an analogous purpose in our setting for the learned velocity field $\widetilde{V}$ and is introduced to transfer stability from the continuous flow to its discrete approximation. Furthermore, Assumption 10 can be relaxed in the temporal direction by requiring only that the accumulated discrete-time sum of the score gradients remains bounded. Alternatively, one may replace this stability condition with trajectory-level approximation assumptions, such as the $L^2$ score-approximation condition H2 in Gentiloni-Silveri & Ocello (2025), which follows a different analytical route; see Remark 21 and Remark 22 in Appendix B.6.

Such Lipschitz control over $\widetilde{V}$ is therefore primarily a technical condition for establishing the convergence theory while remaining reasonable in practice. Indeed, the learned velocity field $\widetilde{V}(t_n, x)$ is trained to approximate the true velocity field $V(t_n, x)$ in Assumption 9, which satisfies the required regularity under the Gaussian-tail assumption (Theorem 8). In practice, neural networks may acquire smoothness and controlled growth through training, while Lipschitz regularity can also be explicitly encouraged through architectural constraints such as spectral normalization (Miyato et al., 2018), Lipschitz-constrained parameterization (Gouk et al., 2021) and Stable residual architecture (Behrmann et al., 2019).

Building upon Theorem 8 and adapting the proof of Theorem 2.5 in Dai et al. (2023), we establish the Lipschitz property of the continuous flow $(\overleftarrow{X}_t)_{t \in [0,1]}$. We further show that the discrete flow $(\overleftarrow{Y}_t)_{t \in [0,1]}$ satisfies an analogous Lipschitz property under Assumption 10. Both statements and their proofs are provided in the appendix B.7.

With the Lipschitz properties of the flow established (see Corollary 23 and Corollary 24), we next quantify how these bounds propagate through the discrete dynamics. Building on Proposition 6, the following theorem provides a convergence result in Föllmer flow case. The objective is to identify conditions under which the optimal $\mathcal{O}(\sqrt{d})$ complexity bound can be derived. While the Gaussian-tail Assumption 7 provides one convenient mechanism for verifying the required regularity conditions, it is not the only possible route; similar regularity properties may also arise from alternative change-of-variables constructions. Consequently, the following theorem should be viewed as a conditional convergence result under Assumptions 9–10, rather than an end-to-end guarantee for a specific neural-network training procedure.

**Theorem 11.** *Suppose that the Gaussian-tail Assumption 7, the accuracy and regularity assumptions 9- 10 on the learned velocity field hold. Using the Euler method for the Föllmer flow with uniform step size $h = t_{n+1} - t_n \leq 1$ ensures $\sqrt{M_0}$ convergence between the target data distribution and the generated distribution:*

$$\mathcal{W}_2(\overrightarrow{P}_0, \overleftarrow{Q}_1) \leq \exp\left(\frac{K_1 + K_2 + K_8}{2}\right)\left(\sqrt{3}\left(K_5\sqrt{M_0} + K_9\right)h + 2\epsilon\right). \tag{14}$$

*where $K_1, K_2, \ldots, K_7$ are dimensionless constants defined in Theorem 8, $K_9$ is defined in the proof of Theorem 11, with explicit expressions given in Table 4 of Appendix C; $K_8$ is defined in Assumption 10. Furthermore, with the covariance of base distribution $C = I_d$ in the Assumption 3, $\mathcal{W}_2(\overrightarrow{P}_0, \overleftarrow{Q}_1) = \mathcal{O}(\sqrt{d}\,h + \epsilon)$.*

Proof see Appendix B.8.

**Corollary 12.** *To reach a distribution $\overleftarrow{Q}_1$ such that $\mathcal{W}_2(\overrightarrow{P}_0, \overleftarrow{Q}_1) = \mathcal{O}(\varepsilon_0)$ with uniform step size $h = t_{n+1} - t_n \leq 1$ requires at most:*

$$h = \mathcal{O}\left(\frac{\epsilon_0}{\sqrt{M_0}}\right), \quad N = \frac{1}{h} = \mathcal{O}\left(\frac{\sqrt{M_0}}{\epsilon_0}\right),$$

*and Assumption 9 to hold with $\epsilon = \mathcal{O}(\epsilon_0)$. Furthermore, $N = \mathcal{O}\left(\frac{\sqrt{d}}{\epsilon_0}\right)$ under the Assumption 3 with $C = I_d$.*

By Proposition 6, the accumulation term is dimension-free under Assumption 7, reflecting the fact that error propagation is controlled by Lipschitz change-of-variables estimates. The only explicit dependence on the ambient dimension arises through the local truncation term, which scales as $\sqrt{\mathrm{Tr}(C)}$. Consequently, the complexity bound established in Corollary 12 grows linearly with the square root of the trace of the forward process's covariance operator, independent of dimension, and thus extends naturally to infinite-dimensional generative models. An illustrative case is provided in Appendix B.11, where we consider Bayesian inverse problems in function spaces. Moreover, under the normalized case ($C = I_d$), our $N$ scales like $\sqrt{d}$, which is shown to be optimal in Proposition 6 of (Gao et al., 2025). Since the probabilistic ODE (Prob ODE) (Song et al., 2021; Gao & Zhu, 2025) can be viewed as a time-rescaled Föllmer flow, the result of Corollary 12 also implies that our method improves the computational complexity of the Prob ODE compared to (Wang & Wang, 2024). We will provide a detailed discussion in Appendix B.9. The result of Corollary 12 indicates that our method improves the computational complexity of the Prob ODE, whereas (Wang & Wang, 2024) shows $N = \mathcal{O}\left(\frac{M_0}{\epsilon_0^2}\left(\log\frac{M_2 + \mathrm{Tr}(C)}{\epsilon_0^2}\right)^3\right)$ under the same setting.

We further verified that our method extends to the 1-rectified flow setting (Liu et al., 2023; Rout et al., 2024). In particular, it applies to the interpolation paths used in the flow built by the first step rectification over independent coupling prior to the recursive construction, and retains the same $\mathcal{O}(\sqrt{d})$ complexity stated in Corollary 12. The proof is deferred to Appendix B.10.

### 3.4 Convergence under bounded-support assumption

Real-world data often lie on low-dimensional manifolds, where the distribution is not absolutely continuous with respect to Lebesgue measure in the ambient dimension, and therefore KL bounds may diverge (Pidstrigach, 2022). This motivates the adoption and study of the manifold assumption (De Bortoli, 2022; Yubin et al., 2025), which, under compactness, entails the following bounded-support assumption.

**Assumption 13.** *Suppose distribution $p_0$ has compact support with $\mathrm{Diam}(\mathrm{Supp}(p_0)) \leq R$ for some constant $R > 0$.*

Let $q_\sigma = \exp\left(-\frac{|x|^2}{2\sigma^2}\right) * q_0$, where $q_0$ satisfies the bounded-support Assumption 13. Consider $g(x) = \log q_\sigma(x) + \frac{|x|^2}{2\sigma^2}$, inspired by similar results in (De Bortoli, 2022; Mooney et al., 2025; Wang & Wang, 2024), we have

$$|\nabla g|_\infty \leq \frac{R}{\sigma^2}, \quad \|\nabla^2 g\|_\infty \leq \frac{2R^2}{\sigma^4}. \tag{15}$$

Set $0 = t_0 < t_1 < \cdots < t_N = 1 - \delta$ as the discretization points, where the early stopping (Lyu et al., 2022) coefficient $\delta \ll 1$. By expressing the distribution of the forward process of Föllmer flow equation 4 at stopping time $\delta$ in the form $q_\sigma$, we obtain the correspondences

$$\sigma^2 \longleftrightarrow 1 - (1-\delta)^2, \quad q_0(x) \longleftrightarrow \frac{1}{1-\delta}\bar{p}_0\left(\frac{1}{1-\delta}x\right),$$

where the expression for $q_0$ follows from the change-of-variables formula under the scaling transformation $x \mapsto (1-\delta)x$.

Under early stopping, the bounded-support Assumption 13 provides a practical alternative to the Gaussian-tail Assumption 7. Applying Theorem 8 then yields the following Lipschitz bound for the velocity field.

**Corollary 14.** *Suppose that the bounded-support Assumption 13 holds. Taking $C = I_d$ and $A = (1 - (1 - \delta)^2)I_d$, then for all $t \in [0, 1 - \delta]$,*

$$|V(t, x)| \leq K_0^* + K_2^* t |x|,$$
$$\|\nabla V(t, \cdot)\|_\infty \leq (K_1^* + K_2^*)t,$$
$$|\partial_t V(t, x)| \leq K_5^* |x| + \frac{K_6^*}{\sqrt{1 - t^2}} + K_7^*,$$

*where coefficients are defined in Table 7 of Appendix C.*

The proof parallels the corollary in (Wang & Wang, 2024). Using the Lipschitz bound from Corollary 14, we obtain a bounded-support-version $W_2$ bound by tracking the constants in Theorem 11.

**Theorem 15.** *Suppose that the bounded-support Assumption 13 and the accuracy and regularity Assumptions 9, 10 hold. Take $C = I_d$, $\delta \ll 1$, then we have*

$$\mathcal{W}_2(\overrightarrow{P}_\delta, \overleftarrow{Q}_{1-\delta}) \leq \exp\left(\frac{3R^2}{2\delta^2} + \frac{1}{2\delta} + \frac{K_8}{2}\right)\left(\sqrt{3}\left(K_5^* \sqrt{M_0} + K_9^*\right)h + 2\epsilon\right),$$

*where $K_5^*$ and $K_9^*$ are dimension-free constants, whose explicit forms given in Table 7 of Appendix C, and the constant $K_8$ is defined in Assumption 10.*

With the result in Theorem 15, we can directly compute the complexity bound under the bounded-support assumption with early stopping technique.

**Corollary 16.** *With $R$ and $\delta$ fixed, achieving a distribution $\overleftarrow{Q}_{1-\delta}$ such that $\mathcal{W}_2(\overrightarrow{P}_\delta, \overleftarrow{Q}_{1-\delta}) = \mathcal{O}(\epsilon_0)$ requires at most: $N = \mathcal{O}\left(\frac{\sqrt{d}}{\epsilon_0}\right)$, and Assumption 9 to hold with $\epsilon = \mathcal{O}(\epsilon_0)$.*

Noticing that,

$$\mathcal{W}_2(\overrightarrow{P}_\delta, \overrightarrow{P}_0) \leq \sqrt{\mathbb{E}|\overrightarrow{X}_\delta - \overrightarrow{X}_0|^2} \leq \sqrt{2d\delta},$$

the complexity bound can also be derived with respect to $\overrightarrow{P}_0$. More precisely, we consider the following practical scenario. Now we assume $R^2 = \mathcal{O}(d)$, then optimizing $\delta$ to achieve $\mathcal{W}_2(\overrightarrow{P}_0, \overleftarrow{Q}_{1-\delta}) = \mathcal{O}(\epsilon_0)$ requires at most logarithmic complexity with $\log N = \mathcal{O}\left(\frac{d^3}{\epsilon_0^4}\right)$.

### 3.5 Numerical Illustration

We validate the optimal $\mathcal{O}(\sqrt{d})$ complexity bound in Theorem 11 through controlled numerical experiments. The experiments are designed to verify the theoretical predictions on discretization error and dimension scaling, rather than to assess the end-to-end performance of neural flow models, with the primary goal of understanding how optimal bound arise under non-trivial data assumptions. To this end, we consider a synthetic distribution for which both the Gaussian-tail constants in Assumption 7 and the associated Föllmer velocity field can be computed explicitly. Specifically, the $d$-dimensional target distribution is chosen as the Gaussian mixture

$$p_{\text{data}}(x) = \frac{1}{2}\exp\left(-\frac{|x - a|^2}{2\tau}\right) + \frac{1}{2}\exp\left(-\frac{|x + a|^2}{2\tau}\right), \tag{16}$$

where $a \in \mathbb{R}^d$ and $\tau > 0$. This distribution admits the Gaussian-tail decomposition

$$p_{\text{data}}(x) = \exp\left(-\frac{|x|^2}{2\tau}\right)\exp(h(x)),$$

with

$$h(x) = \log\left(\exp\left(-\frac{|a|^2}{2\tau} + \frac{\langle a, x\rangle}{\tau}\right) + \exp\left(-\frac{|a|^2}{2\tau} - \frac{\langle a, x\rangle}{\tau}\right)\right).$$

A direct calculation shows that $\nabla h(x) = \tanh\left(\frac{\langle a, x\rangle}{\tau}\right)\frac{a}{\tau}$, and hence $\|\nabla h\|_\infty$ remains uniformly bounded provided $|a|_2$ is dimension-independent. Similarly, the Hessian satisfies $\sup_x \|\nabla^2 h(x)\|_2 \leq \frac{|a|_2^2}{\tau^2}$, which implies that both first- and second-order derivatives are uniformly bounded in $d$. Therefore, the Gaussian-tail Assumption 7 holds with dimension-free constants. In the experiments, we fix $a = (1, 0, \ldots, 0)^\top$.

Under this setting, the score function and the corresponding Föllmer velocity field admit closed-form expressions with

$$
\begin{aligned}
V(t, x) &= \frac{1}{t}\left[x + S(t, x)\right] \\
&= \frac{1}{t}\left[x + C\nabla\log\left(\frac{1}{2}\exp\left(-\frac{|x - ta|_{\bar{A}_t}^2}{2}\right) + \frac{1}{2}\exp\left(-\frac{|x + ta|_{\bar{A}_t}^2}{2}\right)\right)\right] \\
&= \frac{1}{t}\left[(I - C\bar{A}_t^{-1})x + tC\bar{A}_t^{-1}a\tanh\left(t \cdot a^\top \bar{A}_t^{-1}x\right)\right] \\
&= \frac{1}{t}\left[\left(1 - \frac{1}{1 - (1 - \tau)t^2}\right)x + \frac{ta}{1 - (1 - \tau)t^2}\tanh\left(\frac{t\,a^\top x}{1 - (1 - \tau)t^2}\right)\right].
\end{aligned}
$$

allowing us to simulate an ideal Föllmer flow sampler. We measure the discretization error via the coupled estimator $\left(\frac{1}{M}\sum_{i=1}^{M}\|X^{(i)} - Y^{(i)}\|^2\right)^{1/2}$, where $X^{(i)}$ denotes samples generated from the reference continuous-time dynamics and $Y^{(i)}$ denotes samples generated from the Euler discretization using the same initialization and coupled randomness, which upper bounds the Wasserstein distance in Theorem 11. We emphasize that this quantity corresponds to a specific computable coupling and is used to assess whether the convergence behavior and dimension scaling predicted by the theory are observed in practice. For completeness, table 5 in Appendix C listing the numerical values of the constants appearing in Theorem 11 for the dimension-scaling experiment, illustrating their dependence on the problem parameters and confirming that they remain independent of the ambient dimension in this setting.

**Time discretization.** We first study the dependence on the time step size $h$ in dimension $d = 50$. The initial particles are sampled from $X_0 \sim \mathcal{N}(0, I_d)$ with $M = 2000$, and we consider step numbers $N \in \{20, 40, 80, 160, 320, 640\}$. Figure 1a shows the error in log–log scale as a function of $h$. The fitted slope is approximately 0.955, which is consistent with the first-order convergence rate predicted by Theorem 11.

**Dimension scaling.** We next examine the dependence on the dimension $d \in \{5, 10, 20, 40, 80, 160, 320, 640\}$. For each $d$, we determine the minimal number of Euler steps $N$ required to achieve a fixed tolerance $\varepsilon_0 = 10^{-2}$, using a reference solution with $N_{\text{ref}} = 50000$. The resulting scaling, shown in Figure 1b, satisfies $N(d) \propto d^{0.472}$, which is in strong agreement with the theoretical $\mathcal{O}(\sqrt{d})$ complexity.

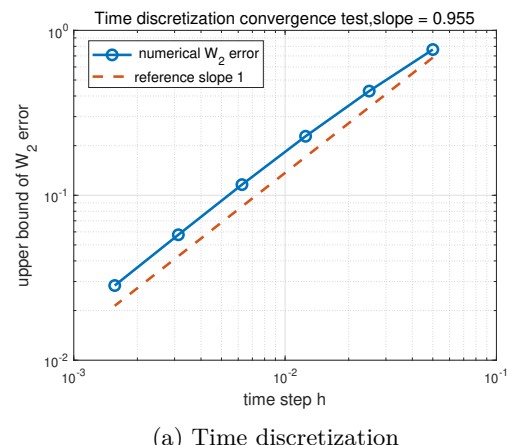

(a) Time discretization

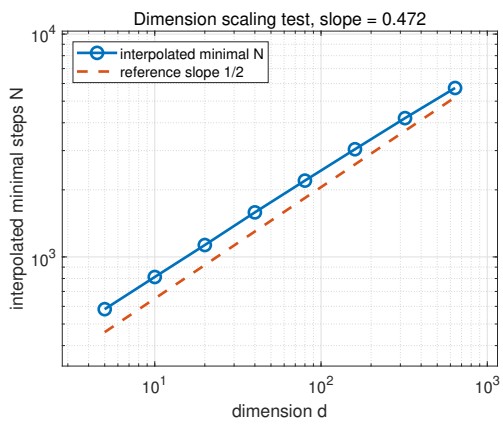

(b) Dimension scaling

## 4 Conclusion and Future Directions

We establish an optimal $\mathcal{O}(\sqrt{\operatorname{Tr} C})$ ($\mathcal{O}(\sqrt{d})$) complexity bound for flow-based generative models under the Wasserstein metric via dimension-free Lipschitz control of truncation errors. The result applies to Föllmer and 1-rectified flows under a Gaussian-tail assumption, extending to bounded-support and infinite-dimensional settings. Numerical results confirm first-order convergence and the predicted scaling.

Several directions merit further investigation:

- **Beyond Gaussian-tails.** Weak log-concavity (Bruno & Sabanis, 2025; Gentiloni-Silveri & Ocello, 2025) extends the Gaussian-tail assumption, but current results only achieve $\mathcal{O}(d)$ complexity. Whether the optimal $\mathcal{O}(\sqrt{d})$ rate persists under this broader condition remains open.

- **Higher-order integrators.** Extending the analysis to higher-order schemes requires refined regularity assumptions (e.g., higher-order time derivatives). Identifying minimal conditions for higher-order Wasserstein convergence is an important direction.

- **Error control and algorithmic design.** Understanding how training objectives, temporal reweighting, and adaptive sampling interact with step-size design (especially under the singular behavior $\frac{1}{\sqrt{1-t^2}}$), and developing data-driven estimates of Lipschitz constants, may lead to sharper Wasserstein error bounds and improved practical performance.

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

## Appendix

## A    Comparison of Lipschitz Bounds via Heat Flow

In this appendix section, we summarize several related results from the literature and compare them with the estimate obtained in this work.

Table 2: Lip changes of variables via Heat flow

| Target $P_0$ | Lip-constant | Result |
|---|---|---|
| log-concave+sym.Ass. | 1+sym.Ass. | Kim & Milman (2012) |
| $\kappa$-log-concave+osc $\leq c$ | $2(2\kappa)^{e^c}$ | Neeman (2022) Thm. 1.3 |
| $\kappa$-log-concave | $e^{\frac{1-\kappa D^2}{2}} D$ | Mikulincer & Shenfeld (2023) Thm. 1 |
| $\kappa$-log-concave + $L$-log-Lipschitz perturbation | $\frac{1}{\sqrt{\kappa}} e^{\left(\frac{L^2}{2\kappa}+2\frac{L}{\sqrt{\kappa}}\right)}$ | Brigati & Pedrotti (2024) Thm. 1.4 |
| **G-tail** Ass. 7 | $\frac{1}{\sqrt{\kappa}} e^{\left(\frac{L^2+L_1}{2\kappa}\right)}$ | This work Cor. 17 |

where $\kappa, c, D, L, L_1, K_1, K_2$ are dimension-independent constant.

**Corollary 17** (This work Cor. 23 with $A = (\kappa I_d)^{-1}$ and $C = I_d$). *Let $\mu = e^{-(W(x)-H(x))}$ be a probability density on $\mathbb{R}^d$ such that $W(x) = \frac{\kappa|x|^2}{2}$ while $H(x)$ being $L$-Lipschitz and $\|\nabla^2 H\|_\infty < L_1$ for some $L, L_1 \geq 0$. Then, there exists a map $T^{\text{flow}} : \mathbb{R}^d \to \mathbb{R}^d$ such that $(T^{\text{flow}})_{\#}\gamma_d = \mu$ and $T^{\text{flow}}$ is $\frac{1}{\sqrt{\kappa}} \exp\left(\frac{L^2+L_1}{2\kappa}\right)$-Lipschitz.*

*Sketch of calculation.* Applying Cor. 23 with $A = (\kappa I_d)^{-1}$ and $C = I_d$, Both $A$ and $C$ are diagonalizable, which leads to a more refined Lip upper bound:

$$\exp\left(\int_0^1 \frac{\|A\|^2(L^2+L_1)t}{(\|A\|t^2+\|C\|(1-t^2))^2}\,\mathrm{d}t + \int_0^1 \frac{\|A-C\|t}{\|A\|t^2+\|C\|(1-t^2)}\,\mathrm{d}t\right)$$
$$\leq \exp\left(\frac{\|A\|(L^2+L_1)}{2\|C\|} + \frac{1}{2}\ln\frac{\|A\|}{\|C\|}\right) \leq \frac{1}{\sqrt{\kappa}}\exp\left(\frac{L^2+L_1}{2\kappa}\right).$$

$\square$

## B    Proofs

In this part, we provide the detailed proofs of the theories in this paper.

### B.1    Proof of Proposition 6

*Proof.* Recall the $p$-Wasserstein distance

$$\mathcal{W}_p(\mu,\nu) := \left(\inf_{\gamma\in\Gamma(\mu,\nu)} \mathbb{E}_{(x,y)\sim\gamma}[\|x-y\|^p]\right)^{1/p},$$

where $\Gamma(\mu,\nu)$ denotes the set of couplings with marginals $\mu$ and $\nu$.

By the push-forward relation equation 3, pushing the coupling $\overleftarrow{Q}_{t_n}$ and $\overleftarrow{P}_{t_n}$ forward through the transport maps $\widetilde{T}_n$ and $T_n$ yields a coupling of $\overleftarrow{Q}_{t_{n+1}}$ and $\overleftarrow{P}_{t_{n+1}}$, which gives

$$
\begin{aligned}
&\mathcal{W}_2(\overleftarrow{P}_{t_{n+1}}, \overleftarrow{Q}_{t_{n+1}}) \\
&\leq \sqrt{\mathbb{E}_{(\overleftarrow{Y}_{t_n}, \overleftarrow{X}_{t_n}) \sim \Gamma(\overleftarrow{Q}_{t_n}, \overleftarrow{P}_{t_n})} |\widetilde{T}_n(\overleftarrow{Y}_{t_n}) - T_n(\overleftarrow{X}_{t_n})|^2} \\
&\leq \sqrt{\mathbb{E}_{(\overleftarrow{Y}_{t_n}, \overleftarrow{X}_{t_n}) \sim \Gamma(\overleftarrow{Q}_{t_n}, \overleftarrow{P}_{t_n})} |\widetilde{T}_n(\overleftarrow{Y}_{t_n}) - \widetilde{T}_n(\overleftarrow{X}_{t_n})|^2} + \sqrt{\mathbb{E}_{\overleftarrow{X}_{t_n} \sim \overleftarrow{P}_{t_n}} |\widetilde{T}_n(\overleftarrow{X}_{t_n}) - T_n(\overleftarrow{X}_{t_n})|^2} \\
&\leq \mathrm{Lip}(\widetilde{T}_n) \sqrt{\mathbb{E}_{(\overleftarrow{Y}_{t_{n-1}}, \overleftarrow{X}_{t_{n-1}}) \sim \Gamma(\overleftarrow{Q}_{t_{n-1}}, \overleftarrow{P}_{t_{n-1}})} |\widetilde{T}_{n-1}(\overleftarrow{Y}_{t_{n-1}}) - \widetilde{T}_{n-1}(\overleftarrow{X}_{t_{n-1}})|^2} \\
&\quad + h\left( \left( \overline{K}\sqrt{M_0} + \frac{\overline{K_1}}{\sqrt{1 - t_n^2}} + \overline{K_2} \right) h + \epsilon \right),
\end{aligned}
$$

where the last inequality uses Assumption 4 (ii) and Assumption 5. Then applying the discrete Grönwall inequality yields equation 11. $\qquad\square$

## B.2 Finite moments under the Gaussian-tail assumption

**Proposition 18.** *Assumption 7 implies that the target distribution admits finite moments of all orders, i.e.,*

$$
\mathbb{E}_{\bar{p}_0}|X|^q < \infty, \qquad \forall q \geq 1. \tag{17}
$$

*Proof.* Since $|\sqrt{C}\nabla h|_\infty < \infty$ and $C$ is positive definite, there exists a constant $L_h < \infty$ such that

$$
|\nabla h(x)| \leq L_h, \qquad x \in \mathbb{R}^d.
$$

By the mean value theorem,

$$
h(x) - h(0) = \int_0^1 \nabla h(tx) \cdot x\, \mathrm{d}t \leq L_h|x|,
$$

which implies

$$
h(x) \leq h(0) + L_h|x|.
$$

On the other hand, since $A$ is symmetric positive definite,

$$
|x|_A^2 = x^\top A^{-1} x \geq \frac{|x|^2}{\|A\|}.
$$

Therefore, using the Gaussian-tail decomposition,

$$
\bar{p}_0(x) = \exp\left( -\frac{|x|_A^2}{2} \right) \exp(h(x)) \leq \exp(h(0)) \exp\left( -\frac{|x|^2}{2\|A\|} + L_h|x| \right).
$$

Completing the square yields

$$
-\frac{|x|^2}{2\|A\|} + L_h|x| = -\frac{1}{2\|A\|}\left( |x| - \|A\|L_h \right)^2 + \frac{\|A\|L_h^2}{2},
$$

and thus

$$
\bar{p}_0(x) \leq C_h \exp\left( -\frac{1}{2\|A\|}\left( |x| - \|A\|L_h \right)^2 \right),
$$

where

$$C_h = \exp\left(h(0) + \frac{\|A\|L_h^2}{2}\right).$$

Let $R := 2\|A\|L_h$. For $|x| \geq R$, we have

$$|x| - \|A\|L_h \geq \frac{|x|}{2},$$

and thus

$$\bar{p}_0(x) \leq C_h \exp\left(-\frac{|x|^2}{8\|A\|}\right), \qquad |x| \geq R.$$

For every $q \geq 1$, decompose

$$\int_{\mathbb{R}^d} |x|^q \bar{p}_0(x)\,\mathrm{d}x = \int_{|x|\leq R} |x|^q \bar{p}_0(x)\,\mathrm{d}x + \int_{|x|>R} |x|^q \bar{p}_0(x)\,\mathrm{d}x.$$

The first term is bounded by

$$\int_{|x|\leq R} |x|^q \bar{p}_0(x)\,\mathrm{d}x \leq R^q \int_{|x|\leq R} \bar{p}_0(x)\,\mathrm{d}x \leq R^q.$$

For the second term, using the Gaussian bound above,

$$\int_{|x|>R} |x|^q \bar{p}_0(x)\,\mathrm{d}x \leq C_h \int_{\mathbb{R}^d} |x|^q \exp\left(-\frac{|x|^2}{8\|A\|}\right)\,\mathrm{d}x < \infty.$$

Therefore,

$$\int_{\mathbb{R}^d} |x|^q \bar{p}_0(x)\,\mathrm{d}x < \infty, \qquad \forall q \geq 1.$$

Consequently, the third-moment condition follows automatically from Assumption 7 and need not be imposed separately. $\square$

### B.3 Well-posedness of Föllmer flow

Under the preceding assumptions and analysis, we establish the well-posedness of the Föllmer model $(\overleftarrow{X}_t)_{t\in[0,1]}$ in the following lemma.

**Lemma 19** (Well-posedness). *Suppose that the Gaussian-tail Assumption 7 hold. Then the Föllmer velocity field is well-defined at the $t = 0$, in the sense that*

$$V(0,x) := \lim_{t\to 0} V(t,x) = \lim_{t\to 0} \frac{x + S(t,x)}{t} = \sqrt{C}\,\mathbb{E}_{\bar{p}_0}[X]. \tag{18}$$

*Consequently, the Föllmer flow $(\overleftarrow{X}_t)_{t\in[0,1]}$ is a unique solution to IVP equation 7. Moreover, the push-forward measure satisfies*

$$\gamma_C \circ (\overleftarrow{X}_1)^{-1} = \bar{p}_0.$$

*Proof.* First, we prove the velocity field $V(t,x)$ is well-defined at $t = 0$ (equation 18 in Lemma 19), i.e.

$$V(0,x) := \lim_{t\to 0} V(t,x) = \lim_{t\to 0} \frac{x + S(t,x)}{t} = \sqrt{C}\,\mathbb{E}_{\bar{p}_0}[X].$$

Let $t \to 0$, then it yields

$$\lim_{t\to 0} V(t,x) = \lim_{t\to 0} \partial_t S(t,x) = \lim_{t\to 0}\left(\frac{C\nabla(\partial_t p_t(x))}{p_t(x)} - \frac{\partial_t p_t(x)}{p_t(x)}S(t,x)\right).$$

By simple calculation, it holds that

$$
\frac{\nabla(\partial_t p_t(x))}{p_t(x)}
$$

$$
= -\partial_t\left(\det B(t)\right)(2\det B(t))^{-1}d\left(-K(t)^T K(t)\left(CB(t)\right)^{-1}-\bar{A}_t^{-1}\right)(x^T x)x
$$

$$
-\partial_t B(t)\left(2B(t)\right)^{-1}K(t)(\sqrt{C}B(t))^{-1}\int_{\mathbb{R}^d} yp(1,y|t,x)\,\mathrm{d}y
$$

$$
+\left(-K(t)\partial_t K(t)(CB(t))^{-1}-K(t)^T K(t)\partial_t\left(B(t)^{-1}\right)\left(2CB(t)\right)^{-1}-\frac{1}{2}\partial_t(\bar{A}_t^{-1})\right)
$$

$$
\cdot\left(2x+\left(-K(t)^T K(t)(CB(t))^{-1}-\bar{A}_t^{-1}\right)(x^T x)x\right.
$$

$$
\left.+2K(t)\left(\sqrt{C}B(t)\right)^{-1}\int_{\mathbb{R}^d}(x^T y)xp(1,y|t,x)\,\mathrm{d}y\right)
$$

$$
+\left(\partial_t K(t)\left(\sqrt{C}B(t)\right)^{-1}+K(t)\partial_t(B(t)^{-1})(\sqrt{C})^{-1}\right)
$$

$$
\cdot\left(\int_{\mathbb{R}^d}\left(y+K(t)(\sqrt{C}B(t))^{-1}(x^T y)y\right)p(1,y|t,x)\,\mathrm{d}y\right.
$$

$$
\left.+\int_{\mathbb{R}^d}\left(-K(t)^T K(t)(CB(t))^{-1}-\bar{A}_t^{-1}\right)(x^T y)xp(1,y|t,x)\,\mathrm{d}y\right)
$$

$$
-\frac{1}{2}\partial_t\left(B(t)^{-1}\right)\int_{\mathbb{R}^d}\left(\left(-K(t)^T K(t)\left(CB(t)\right)^{-1}-\bar{A}_t^{-1}\right)(y^T y)x\right)p(1,y|t,x)\,\mathrm{d}y
$$

$$
-\frac{1}{2}\partial_t\left(B(t)^{-1}\right)\int_{\mathbb{R}^d}K(t)\left(\sqrt{C}B(t)\right)^{-1}(y^T y)yp(1,y|t,x)\,\mathrm{d}y,
$$

where the last term is controlled by bounded third moment (implied by the Gaussian-tail Assumption 7), i.e.

$$
\lim_{t\to0}\int_{\mathbb{R}^d}|y|^3 p(1,y|t,x)\,\mathrm{d}y=\int_{\mathbb{R}^d}|y|^3\lim_{t\to0}p(1,y|t,x)\,\mathrm{d}y=\mathbb{E}_{\bar{p}_0}[|X|^3]<+\infty.
$$

Also

$$
\frac{\partial_t p_t(x)}{p_t(x)}
$$

$$
= -\partial_t\left(\det B(t)\right)\left(2\det B(t)\right)^{-1}d
$$

$$
+x^T\left(-K(t)\partial_t K(t)\left(CB(t)\right)^{-1}-(2C)^{-1}K(t)K(t)^T\partial_t\left(B(t)^{-1}\right)-\frac{1}{2}\partial_t(\bar{A}_t^{-1})\right)x
$$

$$
+\int_{\mathbb{R}^d}x^T\left(\partial_t K(t)\left(\sqrt{C}B(t)\right)^{-1}+K(t)(\sqrt{C})^{-1}\partial_t\left(B(t)^{-1}\right)\right)yp(1,y|t,x)\,\mathrm{d}y
$$

$$
-\int_{\mathbb{R}^d}y^T\frac{1}{2}\partial_t\left(B(t)^{-1}\right)yp(1,y|t,x)\,\mathrm{d}y.
$$

From observing that

$$
\lim_{t\to0}\partial_t\left(\det B(t)\right)=0,\ \lim_{t\to0}\partial_t B(t)=0,\ \lim_{t\to0}K(t)=0,\ \lim_{t\to0}\partial_t(\bar{A}_t^{-1})=0,
$$

we have

$$
\lim_{t\to0}\frac{\partial_t p_t(x)}{p_t(x)}S(t,x)=-\frac{x^\top x}{\sqrt{C}}\mathbb{E}_{\bar{p}_0}[X],\quad \lim_{t\to0}\frac{C\nabla(\partial_t p_t(x))}{p_t(x)}=\sqrt{C}\mathbb{E}_{\bar{p}_0}[X]-\frac{x^\top x}{\sqrt{C}}\mathbb{E}_{\bar{p}_0}[X].
$$

Therefore, it yields $\lim_{t\to0}V(t,x)=\sqrt{C}\mathbb{E}_{\bar{p}_0}[X]$, which completes the proof of equation 18.

Next, together with the regularity of the velocity field (Theorem 8), the Arzelà-Ascoli theorem (Arzela, 1895) ensures the existence of a subsequence $\{V(t_k, \cdot)\}_{k \in N}$ with $t_k \to 0$ that converges locally uniformly to $V(0, \cdot)$, thereby guaranteeing the well-posedness of the ODE equation 7 on the entire time interval $t \in [0, 1]$. $\qquad \square$

### B.4 Proof of Theorem 8

*Proof.* Throughout the proof, we explicitly track the dependence of all constants on the Gaussian-tail quantities in Assumption 7. In particular, the constants $K$, $K_i$ $(i = 0, \ldots, 7)$, and $C_i$ $(i = 1, \ldots, 4)$ depend only on the compatibility quantities associated with $A$ and $C$, together with $|\sqrt{C}\nabla h|_\infty$ and $\|C\nabla^2 h\|_\infty$, and are therefore independent of the ambient dimension $d$; see Table 4 for their explicit expressions.

For simplicity, denote

$$
\begin{aligned}
G(t, x, y) &:= \exp\left( -\frac{\left| K(t)(\sqrt{C})^{-1}x - y \right|^2_{B(t)}}{2} \right), \\
g(t, x, y) &:= \partial_t \left( -\frac{\left| K(t)(\sqrt{C})^{-1}x - y \right|^2_{B(t)}}{2} \right).
\end{aligned}
\tag{19}
$$

Under the Gaussian-tail Assumption 7, the score function of Föllmer flow can be calculated by

$$
\begin{aligned}
S(t, x) &:= C\nabla \log p_t \\
&= C\nabla \log \int_{\mathbb{R}^d} \left( 2\pi \det(B(t)) \right)^{-\frac{d}{2}} G(t, x, y) \cdot \exp\left( -\frac{|x|^2_{\overline{A}_t}}{2} \right) \exp\left( h(\sqrt{C}y) \right) \, \mathrm{d}y,
\end{aligned}
\tag{20}
$$

where $\overline{A}_t = At^2 + C(1 - t^2)$, $K(t) = (A\overline{A}_t^{-1})t$, $B(t) = (A\overline{A}_t^{-1})(1 - t^2)$.

First, we consider the modified score function over the time interval $(0, 1]$,

$$
\begin{aligned}
\widetilde{S}(t, x) &= S(t, x) + C\overline{A}_t^{-1}x \\
&= C\nabla \log \int_{\mathbb{R}^d} \left( 2\pi B(t) \right)^{-\frac{d}{2}} G(t, x, y) \exp\left( h(\sqrt{C}y) \right) \, \mathrm{d}y \\
&= C\frac{\int_{\mathbb{R}^d} \nabla_x G(t, x, y) \exp\left( h(\sqrt{C}y) \right) \, \mathrm{d}y}{\int_{\mathbb{R}^d} G(t, x, y) \exp\left( h(\sqrt{C}y) \right) \, \mathrm{d}y} \\
&= -\frac{K(t)\sqrt{C} \int_{\mathbb{R}^d} \nabla_y G(t, x, y) \exp\left( h(\sqrt{C}y) \right) \, \mathrm{d}y}{\int_{\mathbb{R}^d} G(t, x, y) \exp\left( h(\sqrt{C}y) \right) \, \mathrm{d}y} \\
&= \frac{K(t)\sqrt{C} \int_{\mathbb{R}^d} G(t, x, y) \nabla_y h(\sqrt{C}y) \exp\left( h(\sqrt{C}y) \right) \, \mathrm{d}y}{\int_{\mathbb{R}^d} G(t, x, y) \exp\left( h(\sqrt{C}y) \right) \, \mathrm{d}y},
\end{aligned}
\tag{21}
$$

Here, the last equal sign is derived from integration by parts.

Since $G(t, x, y) \exp\left( h(\sqrt{C}y) \right) \geq 0$,

$$
|\widetilde{S}(t, x)| \leq |K(t)\sqrt{C}\nabla h(\sqrt{C}x)|.
$$

Let $K = \sup_{0 \leq t \leq 1} |\frac{1}{t}K(t)| = \sup_{0 \leq t \leq 1} |A\overline{A}_t^{-1}| \leq \max\{1, \|AC^{-1}\|\}$, we have

$$
|\widetilde{S}(t, \cdot)| \leq K\|C\|^{1/2}|\sqrt{C}\nabla h|_\infty t = K_0 t.
$$

Taking the derivative twice along that direction and using the same method as above, we get :

$$\|\nabla \widetilde{S}(t, \cdot)\|_\infty \le K(t)^2 (\|C\nabla^2 h\|_\infty + |\sqrt{C}\nabla h|_\infty^2) = K_1 t^2.$$

where $K_1 := K^2(\|C\nabla^2 h\|_\infty + |\sqrt{C}\nabla h|_\infty^2)$.

Define $K_2 := \sup_{0 \le t \le 1} \|\frac{1}{t^2}(I - C\overline{A}_t^{-1})\| = \sup_{0 \le t \le 1} \|(A-C)(At^2 + C(1-t^2))^{-1}\|$, then $\|(I - C\overline{A}_t^{-1})\| \le K_2 t^2$.

Recall definition of $V(t, x)$ in equation 7, we have

$$|V(t, x)| = \left| \frac{x + S(t, \cdot)}{t} \right| = \left| \frac{\widetilde{S}(t, \cdot) + (I - C\overline{A}_t^{-1})x}{t} \right| \le K_0 + K_2 t|x|,$$

which implies the velocity field $|V(t, x)|$ remains locally uniformly bounded and grows at most linearly.

Furthermore,

$$\|\nabla V(t, x)\|_\infty = \left\| \nabla \left( \frac{\widetilde{S}(t, x) + (I - C\overline{A}_t^{-1})x}{t} \right) \right\|_\infty \le (K_1 + K_2)t,$$

which yields a Lipschitz constant that is uniform over space and independent of the dimension, ensuring uniform equicontinuity.

Next, we give the properties of $V(t, x)$ with respect to time over $(0, 1)$. By taking equation 21, we have

$$
\begin{aligned}
&\partial_t V(t, x) \\
=&\partial_t \left( \frac{\widetilde{S}(t, \cdot) + (I - C\overline{A}_t^{-1})x}{t} \right) \\
=&\partial_t \left( \frac{K(t)\sqrt{C} \int_{\mathbb{R}^d} G(t, x, y)\nabla_y h(\sqrt{C}y) \exp\left(h(\sqrt{C}y)\right) \, \mathrm{d}y}{t \int_{\mathbb{R}^d} G(t, x, y) \exp\left(h(\sqrt{C}y)\right) \, \mathrm{d}y} \right) + \partial_t \left( \frac{(I - C\overline{A}_t^{-1})x}{t} \right) \\
:=&A_1 + A_2
\end{aligned}
\tag{22}
$$

We begin by calculating the first part,

$$
\begin{aligned}
A_1 =&\partial_t \left( \frac{(A\overline{A}_t^{-1})\sqrt{C} \int_{\mathbb{R}^d} G(t, x, y)\nabla_y h(\sqrt{C}y) \exp\left(h(\sqrt{C}y)\right) \, \mathrm{d}y}{\int_{\mathbb{R}^d} G(t, x, y) \exp\left(h(\sqrt{C}y)\right) \, \mathrm{d}y} \right) \\
=&\frac{\partial_t(A\overline{A}_t^{-1})\sqrt{C} \int_{\mathbb{R}^d} G(t, x, y)\nabla_y h(\sqrt{C}y) \exp\left(h(\sqrt{C}y)\right) \, \mathrm{d}y}{\int_{\mathbb{R}^d} G(t, x, y) \exp\left(h(\sqrt{C}y)\right) \, \mathrm{d}y} \\
&+ \frac{A\overline{A}_t^{-1}\sqrt{C} \int_{\mathbb{R}^d} G(t, x, y)g(t, x, y) \exp\left(h(\sqrt{C}y)\right) \nabla_y h(\sqrt{C}y) \, \mathrm{d}y}{\int_{\mathbb{R}^d} G(t, x, y) \exp\left(h(\sqrt{C}y)\right) \, \mathrm{d}y} \\
&- \frac{A\overline{A}_t^{-1}\sqrt{C} \int_{\mathbb{R}^d} G(t, x, y) \exp\left(h(\sqrt{C}y)\right) \nabla_y h(\sqrt{C}y) \, \mathrm{d}y}{\int_{\mathbb{R}^d} G(t, x, y) \exp\left(h(\sqrt{C}y)\right) \, \mathrm{d}y} \\
&\cdot \frac{\int_{\mathbb{R}^d} G(t, x, y)g(t, x, y) \exp\left(h(\sqrt{C}y)\right) \, \mathrm{d}y}{\int_{\mathbb{R}^d} G(t, x, y) \exp\left(h(\sqrt{C}y)\right) \, \mathrm{d}y}
\end{aligned}
\tag{23}
$$

$$\leq \frac{\partial_t(A\bar{A}_t^{-1})\sqrt{C}\displaystyle\int_{\mathbb{R}^d} G(t,x,y)\nabla_y h(\sqrt{C}y)\exp\left(h(\sqrt{C}y)\right)\,\mathrm{d}y}{\displaystyle\int_{\mathbb{R}^d} G(t,x,y)\exp\left(h(\sqrt{C}y)\right)\,\mathrm{d}y}$$

$$+ A\bar{A}_t^{-1}\sqrt{C}\,\mathrm{Cov}_{p_t(y|x)}\left(g(t,x,y),\nabla_y h(\sqrt{C}y)\right)$$

$$\leq \frac{\partial_t(A\bar{A}_t^{-1})\sqrt{C}\displaystyle\int_{\mathbb{R}^d} G(t,x,y)\nabla_y h(\sqrt{C}y)\exp\left(h(\sqrt{C}y)\right)\,\mathrm{d}y}{\displaystyle\int_{\mathbb{R}^d} G(t,x,y)\exp\left(h(\sqrt{C}y)\right)\,\mathrm{d}y}$$

$$+ A\bar{A}_t^{-1}\sqrt{C}\sqrt{\mathrm{Var}\left(g(t,x,y)\right)}\sqrt{\mathrm{Var}\left(\nabla_y h(\sqrt{C}y)\right)}$$

$$:=I_1 + I_2,$$

where

$$p_t(y|x) = \frac{G(t,x,y)\exp\left(h(\sqrt{C}y)\right)}{\displaystyle\int_{\mathbb{R}^d} G(t,x,y)\exp\left(h(\sqrt{C}y)\right)\,\mathrm{d}y}.$$

Define $K_3 := \sup_{0\leq t\leq 1}\|\frac{A\partial_t(\bar{A}_t^{-1})}{t}\| = \sup_{0\leq t\leq 1}\|2A(A-C)((At^2 + C(1-t^2))^2)^{-1}\|$, we obtain the first term of $A_1$,

$$\begin{aligned}
|I_1| &\leq \left\|2A(A-C)\left((At^2 + C(1-t^2))^2\right)^{-1}t\right\||\sqrt{C}||\sqrt{C}\nabla h|_\infty \\
&\leq |\sqrt{C}||\sqrt{C}\nabla h|_\infty K_3 t \\
&\leq K_0 K_3 K^{-1}.
\end{aligned} \tag{24}$$

For the second term $I_2$ in equation 23, the analysis is carried out separately for the following two cases:

**Case I -** $\|B(t)\| \leq \frac{1}{2\|C\nabla^2 h\|_\infty}$**:**  Then $p_t(y|x)$ is a log-concave measure as

$$-\nabla^2 \log p_t(y|x) = B(t)^{-1} - C\nabla^2 h \succ 0. \tag{25}$$

Then, by Brascamp-Lieb inequality, we have

$$\sqrt{\mathrm{Var}\left(g(t,x,y)\right)} \leq \sqrt{\mathbb{E}_{p_t(y|x)}\left|\nabla^T g(t,x,y)\left(-\nabla^2 \log p_t(y|x)\right)^{-1}\nabla g(t,x,y)\right|}, \tag{26}$$

here $g(t,x,y)$ is defined in equation 19 and its gradient is given by

$$\nabla g(t,x,y) = \partial_t K(t)B(t)^{-1}(\sqrt{C})^{-1}x - \partial_t B(t)B(t)^{-2}\left(K(t)(\sqrt{C})^{-1}x - y\right). \tag{27}$$

Recall that $K(t) = (A\bar{A}_t^{-1})t$, $B(t) = (A\bar{A}_t^{-1})(1-t^2)$.

Substituting equation 27 into equation 26 yields

RHS of equation 26

$$
= \left( \int_{\mathbb{R}^d} \left| \left( \partial_t K(t) B(t)^{-1} (\sqrt{C})^{-1} x - \partial_t B(t) B(t)^{-2} \left( K(t) (\sqrt{C})^{-1} x - y \right) \right)^T \right. \right.
$$
$$
\cdot \left( B(t)^{-1} - C \nabla^2 h \right)^{-1}
$$
$$
\left. \left. \cdot \left( \partial_t K(t) B(t)^{-1} (\sqrt{C})^{-1} x - \partial_t B(t) B(t)^{-2} \left( K(t) (\sqrt{C})^{-1} x - y \right) \right) p_t(y|x) \right| dy \right)^{\frac{1}{2}}
$$

$$
\leq \left( \int_{\mathbb{R}^d} \left| \left( \partial_t K(t) B(t)^{-1} C^{-1} x \right)^T \left( I - B(t) C \nabla^2 h \right)^{-1} \partial_t K(t) x \, p_t(y|x) \right| dy \right.
$$
$$
+ \int_{\mathbb{R}^d} \left| \left( -2 \partial_t K(t) B(t)^{-1} (\sqrt{C})^{-1} \partial_t B(t) x \right)^T \left( I - B(t) C \nabla^2 h \right)^{-1} \right.
$$
$$
\cdot \left( \left( K(t) (\sqrt{C})^{-1} x - y \right) B(t)^{-1} \right) p_t(y|x) \right| dy
$$
$$
+ \int_{\mathbb{R}^d} \left| \left( K(t) (\sqrt{C})^{-1} x - y \right)^T B(t)^{-2} \left( \partial_t B(t) \right)^2 \left( I - B(t) C \nabla^2 h \right)^{-1} \right.
$$
$$
\left. \left. \cdot B(t)^{-1} \left( K(t) (\sqrt{C})^{-1} x - y \right) p_t(y|x) \right| dy \right)^{\frac{1}{2}}.
$$

Applying the triangle inequality $|a + b + c|^{\frac{1}{2}} \leq |a|^{\frac{1}{2}} + |b|^{\frac{1}{2}} + |c|^{\frac{1}{2}}$, we obtain:

RHS of equation 26

$$
\leq \left( \int_{\mathbb{R}^d} \left| \left( \partial_t K(t) B(t)^{-1} C^{-1} x \right)^T \left( I - B(t) C \nabla^2 h \right)^{-1} \partial_t K(t) x \, p_t(y|x) \right| dy \right)^{\frac{1}{2}}
$$
$$
+ \left( \int_{\mathbb{R}^d} \left| - \left( 2 \partial_t K(t) B(t)^{-1} (\sqrt{C})^{-1} \partial_t B(t) x \right)^T \left( I - B(t) C \nabla^2 h \right)^{-1} \right. \right.
$$
$$
\left. \left. \cdot \left( \left( K(t) (\sqrt{C})^{-1} x - y \right) B(t)^{-1} \right) p_t(y|x) \right| dy \right)^{\frac{1}{2}}
$$
$$
+ \left( \int_{\mathbb{R}^d} \left| \left( K(t) (\sqrt{C})^{-1} x - y \right)^T B(t)^{-2} (\partial_t B(t))^2 \right. \right.
$$
$$
\left. \left. \cdot \left( I - B(t) C \nabla^2 h \right)^{-1} B(t)^{-1} \left( K(t) (\sqrt{C})^{-1} x - y \right) p_t(y|x) \right| dy \right)^{\frac{1}{2}}
$$
$$
:= R_1 + R_2 + R_3.
$$

By a straightforward estimation,

$$
R_1 \leq |\partial_t K(t)| \|\sqrt{C}^{-1}\| \sqrt{\|B(t)\|^{-1}} \sqrt{(1 - \|B(t)\| \|C \nabla^2 h\|)^{-1}} |x|.
$$

It follows from integration by parts that

$$
- \int_{\mathbb{R}^d} \left( K(t) (\sqrt{C})^{-1} x - y) B(t)^{-1} \right) p_t(y|x) \, dy
$$
$$
= - \frac{\int_{\mathbb{R}^d} \nabla_y G(t, x, y) \exp\left( h(\sqrt{C} y) \right) \, dy}{\int_{\mathbb{R}^d} G(t, x, y) \exp\left( h(\sqrt{C} y) \right) \, dy}
$$
$$
\leq |\sqrt{C} \nabla h|_\infty,
$$

which immediately implies

$$R_2 = \left( \frac{1}{\left| \int_{\mathbb{R}^d} G(t,x,y) \exp\left( h(\sqrt{C}y) \right) \, \mathrm{d}y \right|} \int_{\mathbb{R}^d} \left| \left( -2\partial_t K(t)(\sqrt{C})^{-1} x B(t)^{-1} \partial_t B(t) \right. \right. \right.$$
$$\left. \left. \left. \cdot \left( I - B(t) C \nabla^2 h \right)^{-1} \nabla_y G(t,x,y) \exp(h(\sqrt{C}y) \right| \mathrm{d}y \right)^{\frac{1}{2}}$$
$$\leq \sqrt{2 \|\partial_t K(t)\| \|\partial_t B(t)\| \|\nabla h\|_\infty} \sqrt{\|B(t)\|^{-1}} \sqrt{(1 - \|B(t)\| \|C\nabla^2 h\|)^{-1}} \sqrt{|x|}.$$

Similarly, applying integration by parts twice, we obtain

$$R_3 \leq \|\partial_t B(t)\| \sqrt{\|C(|\nabla h|_\infty^2 + \|\nabla^2 h\|_\infty)\|} \sqrt{\|B(t)^{-1}\|} \sqrt{(1 - \|B(t)\| \|C\nabla^2 h\|_\infty)^{-1}}$$
$$+ \|\partial_t B(t)\| \|B(t)^{-1}\| \sqrt{(1 - \|B(t)\| \|C\nabla^2 h\|_\infty)^{-1}}.$$

Then we have

RHS of equation 26
$$\leq \left( |\partial_t K(t) x|^2 \|C\|^{-1} + 2\|\partial_t K(t)\| \|\partial_t B(t)\| \|\nabla h\|_\infty |x| \right.$$
$$\left. + \|\partial_t B(t)\|^2 (\|C\nabla^2 h\|_\infty + |\sqrt{C}\nabla h|_\infty^2) \right)^{\frac{1}{2}} \cdot \sqrt{\left( \|B(t)\| \left( 1 - \|B(t)\| \|C\nabla^2 h\|_\infty \right) \right)^{-1}}$$
$$+ \|\partial_t B(t)\| \sqrt{\left( \|B(t)\|^2 (1 - \|B(t)\| \|C\nabla^2 h\|_\infty) \right)^{-1}}.$$

Combining with

$$\sqrt{\mathrm{Var}\left( \nabla_y h(\sqrt{C}y) \right)} \leq \|C\nabla^2 h\|_\infty \| \sqrt{B(t)} \| \sqrt{(1 - \|B(t)\| \|C\nabla^2 h\|_\infty)^{-1}},$$

we derive that

$$|I_2| \leq K \|C\nabla^2 h\|_\infty (1 - \|B(t)\| \|C\nabla^2 h\|_\infty)^{-1} \left( |\partial_t K(t) x| \right.$$
$$\left. + \sqrt{2\|\partial_t K(t)\| \|\partial_t B(t)\| \|C\nabla h\|_\infty |x|} + \|\partial_t B(t)\| \sqrt{\|C\nabla^2 h\|_\infty + |\sqrt{C}\nabla h|_\infty^2} \right)$$
$$+ \| \sqrt{B(t)^{-1}} \| K \|\sqrt{C}\| \|\partial_t B(t)\| \|C\nabla^2 h\|_\infty (1 - \|B(t)\| \|C\nabla^2 h\|_\infty)^{-1}$$
$$\leq 2K \|C\nabla^2 h\|_\infty \left( |\partial_t K(t) x| + \sqrt{2\|\partial_t K(t)\| \|\partial_t B(t)\| \|\nabla h\|_\infty |x|} \right.$$
$$\left. + \|\partial_t B(t)\| \sqrt{\|C\nabla^2 h\|_\infty + |\sqrt{C}\nabla h|_\infty^2} \right)$$
$$+ 2K \|\sqrt{C}\| \|\partial_t B(t)\| \|C\nabla^2 h\|_\infty \| \sqrt{B(t)^{-1}} \|$$
$$\leq 2K_1 C_1 |x| + 2\sqrt{2} K_1 \sqrt{K_0} \sqrt{C_1} \sqrt{C_2} \sqrt{|x|} + 2K_1^{\frac{3}{2}} C_2 + 2K_1 \|\sqrt{C}\| C_2 \sqrt{\|B(t)\|^{-1}}$$
$$\leq 3K_1 C_1 |x| + \frac{2K^{-1} K_1 \|\sqrt{C}\| C_2}{\sqrt{1 - t^2}} + C_3,$$

where the second inequality use the fact $(1 - \|B(t)\| \|C\nabla^2 h\|_\infty)^{-1} \leq 2$ under Case I, while the last inequality is obtained using Young's inequality, $C_1, C_2, C_3$ are dimension-free constants defined in table 4.

**Case II -** $\|B(t)\| > \frac{1}{2\|C\nabla^2 h\|}$: According to the definition of variance

$$\mathrm{Var}(g(t,x,y)) = \int_{\mathbb{R}^d} \left( g(t,x,y) \right)^2 p_t(y|x) \, \mathrm{d}y - \left( \int_{\mathbb{R}^d} g(t,x,y) p_t(y|x) \, \mathrm{d}y \right)^2$$
$$\leq \int_{\mathbb{R}^d} \left( g(t,x,y) \right)^2 p_t(y|x) \, \mathrm{d}y,$$

We have the following estimate, analogous to the right-hand side of equation 26:

$$
\sqrt{\mathrm{Var}\,(g(t,x,y))}
$$
$$
\leq \frac{\|\sqrt{B(t)^{-1}}\|}{2}\Big(2\|\sqrt{C}^{-1}\||\partial_t K(t)x| + 2\sqrt{3}\sqrt{\|\partial_t K(t)\|\|\partial_t B(t)\|}|\nabla h|_\infty|x|
$$
$$
+ \sqrt{6}\|\partial_t B(t)\|\sqrt{\|C\nabla^2 h\|_\infty + |\sqrt{C}\nabla h|^2_\infty}\Big)
$$
$$
+ |\partial_t K(t)x|\sqrt{\|\nabla^2 h\|_\infty + |\nabla h|^2_\infty} + \frac{1}{2}\|\partial_t B(t)\|\sqrt{|\sqrt{C}\nabla h|_\infty(\|C\nabla^2 h\|_\infty + |\sqrt{C}\nabla h|^2_\infty)}
$$
$$
+ \sqrt{\|\partial_t K(t)\|\|\partial_t B(t)\||\nabla h|_\infty(\|C\nabla^2 h\|_\infty + |\sqrt{C}\nabla h|^2_\infty)|x|} + \frac{\sqrt{3}}{2}\|\partial_t B(t)\|\|B(t)^{-1}\|
$$
$$
\leq \sqrt{\frac{\|\nabla^2 h\|_\infty}{2}}\Big(2|\partial_t K(t)x| + 2\sqrt{3}\sqrt{\|\partial_t K(t)\|\|\partial_t B(t)\|}|C\nabla h|_\infty|x|
$$
$$
+ \sqrt{6}\|\sqrt{C}\|\|\partial_t B(t)\|\sqrt{\|C\nabla^2 h\|_\infty + |\sqrt{C}\nabla h|^2_\infty}\Big) + |\partial_t K(t)x|\sqrt{\|\nabla^2 h\|_\infty + |\nabla h|^2_\infty}
$$
$$
+ \sqrt{\|\partial_t K(t)\|\|\partial_t B(t)\||\nabla h|_\infty(\|C\nabla^2 h\|_\infty + |\sqrt{C}\nabla h|^2_\infty)|x|}
$$
$$
+ \frac{1}{2}\|\partial_t B(t)\|\sqrt{|\sqrt{C}\nabla h|_\infty(\|C\nabla^2 h\|_\infty + |\sqrt{C}\nabla h|^2_\infty)} + \sqrt{3}\|\partial_t B(t)\|\|C\nabla^2 h\|_\infty.
$$

Together with

$$
\sqrt{\mathrm{Var}\left(\nabla_y h(\sqrt{C}y)\right)} \leq |\sqrt{C}\nabla h|_\infty,
$$

we obtain

$$
|I_2| \leq K|\sqrt{C}\nabla h|_\infty\Bigg(|\partial_t K(t)x|\big(\sqrt{2\|C\nabla^2 h\|_\infty} + \sqrt{(\|C\nabla^2 h\|_\infty + |\sqrt{C}\nabla h|^2_\infty)}\big)
$$
$$
+ \sqrt{6}\sqrt{\|C\nabla^2 h\|_\infty}\sqrt{\|\partial_t K(t)\|\|\partial_t B(t)\||C\nabla h|_\infty|x|}
$$
$$
+ \sqrt{\|\partial_t K(t)\|\|\partial_t B(t)\|\sqrt{C}\nabla h|_\infty(\|C\nabla^2 h\|_\infty + |\sqrt{C}\nabla h|^2_\infty)|x|}
$$
$$
+ \sqrt{3}\|\sqrt{C}\|\sqrt{\|C\nabla^2 h\|_\infty}\|\|\partial_t B(t)\|\|\sqrt{\|C\nabla^2 h\|_\infty + |\sqrt{C}\nabla h|^2_\infty}
$$
$$
+ \frac{1}{2}\|\sqrt{C}\|\|\partial_t B(t)\|\sqrt{|\sqrt{C}\nabla h|_\infty(\|C\nabla^2 h\|_\infty + |\sqrt{C}\nabla h|^2_\infty)}
$$
$$
+ \sqrt{3}\|\sqrt{C}\|\|\partial_t B(t)\|\|C\nabla^2 h\|_\infty\Bigg)
$$
$$
\leq (1+\sqrt{2})K_0\sqrt{K_1}C_1|x| + (1+\sqrt{6})K_0^{\frac{3}{2}}\sqrt{K_1}\sqrt{C_1}\sqrt{C_2}|\sqrt{x}|
$$
$$
+ K_0 C_2(2\sqrt{3}K_1 + \frac{1}{2}\sqrt{K_0}\sqrt{K_1})
$$
$$
\leq 2(1+\sqrt{2})K_0\sqrt{K_1}C_1|x| + C_4,
$$

where $C_4$ defined in table 4 is also constant indepent of dimension.

Combining the above two cases, we obtain

$$
|I_2| \leq \max\{3K_1 C_1, 2(1+\sqrt{2})K_0\sqrt{K_1}C_1\}|x| + \frac{2K^{-1}K_1\|\sqrt{C}\|C_2}{\sqrt{1-t^2}} + \max\{C_3, C_4\}. \tag{28}
$$

Using equation 23, equation 24 and equation 28, we derive

$$
|A_1| \leq |I_1| + |I_2|
$$
$$
\leq K_0 K_3 K^{-1} + \max\{3K_1 C_1, 2(1+\sqrt{2})K_0\sqrt{K_1}C_1\}|x|
$$
$$
+ \frac{2K_1 K^{-1}\|\sqrt{C}\|C_2}{\sqrt{1-t^2}} + \max\{C_3, C_4\}.
$$

The next step is to calculate the absolute value of the second term of equation 22, i.e.

$$|A_2| = \left| \frac{-C\partial_t(\overline{A}_t^{-1})xt + (I - C\overline{A}_t^{-1})x}{t^2} \right|$$

$$\leq \|2C(A-C)\big((At^2 + C(1-t^2))^2\big)^{-1}\||x| + K_2|x|$$

$$\leq (K_4 + K_2)|x|.$$

where $K_4 := \sup_{0 \leq t \leq 1} \|\frac{C\partial_t(\overline{A}_t^{-1})}{t}\| = \sup_{0 \leq t \leq 1} \|2C(A-C)\big((At^2 + C(1-t^2))^2\big)^{-1}\|$.

It then follows from equation 22 that

$$|\partial_t V(t,x)| \leq |A_1| + |A_2| \leq K_5|x| + \frac{K_6}{\sqrt{1-t^2}} + K_7.$$

where $K_5 = \max\{3K_1C_1, 2(1+\sqrt{2})K_0\sqrt{K_1}C_1\} + K_2 + K_4$, $K_6 = 2K_1K^{-1}\|\sqrt{C}\|C_2$, $K_7 = \max\{C_3, C_4\} + K_0K_3K^{-1}$.

When $t = 0$, by equation 18 in the subsequently stated well-posedness Lemma 19, we have

$$|V(0,x)| = \left| \sqrt{C}\mathbb{E}_{\overline{p}_0}[X] \right| \lesssim K\|C\|^{1/2} \left| \sqrt{C}\nabla h \right|_\infty = K_0.$$

Similarly, $\|\nabla V(0, \cdot)\|_\infty$ is also bounded by $K_1 + K_2$. Then we conclude that the velocity field $V(t, x)$ satisfied the condition of Theorem 8 for all $t \in [0, 1]$.

For clarity, the coefficients are summarized in Table 4.

In corollary 14, we take $C = I_d$, $A = (1 - (1-\delta)^2)I_d$, $\overline{A}_t = (1 - t^2)I_d$. According to equation 15, we deduce the explicit for coefficients $K_0^*, K_1^*, K_2^*, K_3^*, K_4^*, K_5^*, K_6^*, K_7^*, K_9^*$ in Table 7. □

### B.5 Averaged Föllmer velocity

In this appendix section, we show that the regularity estimate established in Theorem 8 extends naturally to an averaged version of the Föllmer velocity inspired by the averaged-velocity construction in MeanFlow (Geng et al., 2025), preserving the same linear-growth behavior under temporal averaging. Motivated by the averaged-velocity construction in MeanFlow (Geng et al., 2025), we introduce an analogous notion for the Föllmer flow and define the *averaged Föllmer velocity* as

$$\overline{V}(x_t, r, t) := \frac{1}{t-r} \int_r^t V(\tau, x_\tau)\,\mathrm{d}\tau.$$

**Corollary 20.** *Under the regularity condition equation 13 satisfied by the Föllmer velocity field, a direct calculation gives the uniform bound*

$$\left| \overline{V}(x_t, r, t) \right| \leq K_* + \frac{t+r}{2}C_*K_2|x|$$

*with $C_* = \exp\big(\frac{K_2}{2}\big)$ and $K_* = K_0 + \frac{C_*K_0K_2}{6}(2t^2 - tr - r^2)$, demonstrating that the averaged Föllmer velocity preserves the same linear growth property as the original velocity field.*

*Proof.* By the equation 13 in Theorem 8, we have

$$|\overline{V}(x_t, r, t)| \leq \frac{1}{t-r} \int_r^t |V(\tau, x_\tau)|\,\mathrm{d}\tau$$

$$\leq \frac{1}{t-r} \int_r^t \big(K_0 + K_2\,\tau\,|x_\tau|\big)\,\mathrm{d}\tau$$

$$= K_0 + \frac{K_2}{t-r} \int_r^t \tau\,|x_\tau|\,\mathrm{d}\tau. \tag{29}$$

We next control the trajectory. Taking norms in the ODE yields

$$\frac{d}{d\tau}|x_\tau| \le |V(\tau, x_\tau)| \le K_0 + K_2\,\tau\,|x_\tau|.$$

Applying Grönwall's inequality yields

$$|x_\tau| \le \exp\left(\frac{K_2}{2}(\tau^2 - r^2)\right)\left(|x| + K_0\int_r^\tau \exp\left(-\frac{K_2}{2}(s^2 - r^2)\right)\,\mathrm{d}s\right).$$

Since $s \in [r, 1]$, we have $\exp\left(-\frac{K_2}{2}(s^2 - r^2)\right) \le 1$, which implies $\int_r^\tau \exp\left(-\frac{K_2}{2}(s^2 - r^2)\right)\,\mathrm{d}s \le \tau - r$. Substituting this estimate into the previous inequality yields

$$|x_\tau| \le \exp\left(\frac{K_2}{2}(\tau^2 - r^2)\right)\left(|x| + K_0(\tau - r)\right).$$

Using $\tau \le 1$, we further obtain the uniform bound

$$\exp\left(\frac{K_2}{2}(\tau^2 - r^2)\right) \le \exp\left(\frac{K_2}{2}\right) =: C_*,$$

and hence

$$|x_\tau| \le C_*\big(|x| + K_0(\tau - r)\big).$$

Substituting this into equation 29, we obtain

$$\int_r^t \tau|x_\tau|\,d\tau \le C_* \int_r^t \tau\big(|x| + K_0(\tau - r)\big)\,d\tau$$
$$= C_*|x|\int_r^t \tau\,d\tau + C_*K_0 \int_r^t \tau(\tau - r)\,\mathrm{d}\tau.$$

A direct computation yields

$$\int_r^t \tau\,d\tau = \frac{t^2 - r^2}{2}, \quad \int_r^t \tau(\tau - r)\,d\tau = \frac{t^3 - r^3}{3} - r\frac{t^2 - r^2}{2}.$$

Dividing by $(t - r)$ and using $\frac{t^2 - r^2}{t - r} = t + r$, we obtain

$$|\bar{V}(x, r, t)| \le K_0 + C_*\frac{K_2}{2}(t + r)|x| + R(r, t),$$

where $R(r, t) = \frac{C_* K_0 K_2}{6}(2t^2 - tr - r^2)$. Since $r, t \in [0, 1]$, $R(r, t)$ is uniformly bounded and can be absorbed into the constant. Redefining $K_* = K_0 + R(r, t)$ accordingly yields the result

$$|\bar{V}(x, r, t)| \le K_* + \frac{t + r}{2}C_*K_2|x|.$$

$\square$

## B.6 Result under the relaxation of Assumption 10

Assumption 10 is used to control the stability of the learned discrete flow in the Wasserstein error recursion. In this section, we discuss two possible ways to weaken or replace this condition. First, the uniform-in-time Lipschitz bound can be relaxed to an accumulated discrete-time gradient bound (Remark 21). Second, one may instead impose trajectory-level approximation assumptions, such as the $L^2$ score-approximation condition H2 in Gentiloni-Silveri & Ocello (2025), which follows a different analytical route (Remark 22).

**Remark 21** (Relaxation of Assumption 10). *Assumption 3.13 can be relaxed as follows. We only require that for all $k \in \{0, 1, \ldots, N-1\}$,*

$$\sum_{n=0}^{N-1}(t_{n+1} - t_n)\|\nabla\widetilde{V}(t_n, \cdot)\|_\infty \leq B, \tag{30}$$

*for some constant $B > 0$. Under this condition, the Lipschitz bound stated in Corollary 24 is refined to $\exp(B)$. Consequently, Theorem 11 remains valid with the product of Lipschitz coefficient $\prod_{j=0}^{N-1}\mathrm{Lip}(\widetilde{T}_j)$, where the factor $\exp\left(\frac{K_1+K_2+K_8}{2}\right)$ is replaced throughout by $\exp(B)$.*

*Proof.* Denote $\beta_n = \|\nabla\widetilde{V}(t_n, \cdot)\|_\infty$, then

$$\frac{\mathrm{d}t|\overleftarrow{Y}_t(x) - \overleftarrow{Y}_t(y)|}{\mathrm{d}t} \leq |\widetilde{V}\big(t_n, \overleftarrow{Y}_{t_n}(x)\big) - \widetilde{V}\big(t_n, \overleftarrow{Y}_{t_n}(y)\big)| \leq \beta_n|\overleftarrow{Y}_{t_n}(x) - \overleftarrow{Y}_{t_n}(y)|.$$

Integration over time yields

$$|\overleftarrow{Y}_{t_{n+1}}(x) - \overleftarrow{Y}_{t_{n+1}}(y)| \leq |\overleftarrow{Y}_{t_n}(x) - \overleftarrow{Y}_{t_n}(y)| + (t_{n+1} - t_n)\beta_n|\overleftarrow{Y}_{t_n}(x) - \overleftarrow{Y}_{t_n}(y)|,$$

yielding $\mathrm{Lip}(\widetilde{T}_n) \leq 1 + (t_{n+1} - t_n)\beta_n$. Recall equation 30, we obtain

$$\mathrm{Lip}(\overleftarrow{Y}_1(x)) \leq \prod_{j=0}^{N-1}\mathrm{Lip}(\widetilde{T}_j) \leq \exp\left(\sum_{n=0}^{N-1}(t_{n+1} - t_n)\beta_n\right) = \exp(B). \tag{31}$$

Subsequently, we analyze the impact of relaxing Assumption 10 on convergence. The preliminary estimate equation 34 still hold. Under the new Lipschitz coefficient $\mathrm{Lip}(\widetilde{T}_n) \leq 1 + (t_{n+1} - t_n)\beta_n$, equation equation 35 takes the following form,

$$\begin{aligned}
\mathcal{W}_2(\overleftarrow{P}_1, \overleftarrow{Q}_1) &\leq \exp(B)\,\mathcal{W}_2(\overleftarrow{P}_0, \overleftarrow{Q}_0) + \exp(B)\,\frac{\sqrt{3}K_6\pi}{4}h \\
&\quad + \frac{\exp(B)}{h}\cdot h\left(\frac{\sqrt{3}}{2}\left(K_5\sqrt{M_0} + K_7\right)h + \epsilon\right) \\
&\leq \exp(B)\left(\frac{\sqrt{3}}{2}\left(K_5\sqrt{M_0} + K_9\right)h + \epsilon\right).
\end{aligned} \tag{32}$$

where $K_5$ and $K_9$ are same with Theorem 11. $\qquad\square$

*Remark 22* (Alternative trajectory-level approximation assumptions). *Beyond the relaxation above, another possible route is to replace the stability condition in Assumption 10 with trajectory-level approximation assumptions; for example, Gentiloni-Silveri & Ocello (2025) impose the $L^2$ score-approximation condition H2. This condition controls the approximation error along the sampling trajectory and can be related to standard score-matching errors under an additional spatial Lipschitz assumption on the learned score estimator. By contrast, our assumptions are formulated on the forward process, together with the stability condition in Assumption 10, which provides a tractable way to close the Wasserstein recursion. Since score functions are typically learned from the exact marginal distribution equation 5, the trajectory-level approximation error appearing in H2 is generally difficult to verify. In contrast, Assumption 10 yields a theoretically tractable framework, and our numerical results support the predicted $O(\sqrt{d})$ scaling in the regime where the score approximation is sufficiently accurate, improving upon the $O(d)$ dependence obtained in Gentiloni-Silveri & Ocello (2025). Whether such Lipschitz control over $\widetilde{V}$ is intrinsically necessary for achieving a $\sqrt{d}$- type optimal complexity bound remains an open question.*

### B.7 Lipschitzness of Föllmer flow

Under Assumption 7, we establish the Lipschitz property of the continuous flow $(\overleftarrow{X}_t)_{t\in[0,1]}$.

**Corollary 23** (Lipschitzness of continuous flow). *If $\bar{p}_0$ follows the Gaussian-tail Assumption 7, then the Föllmer flow $(\overleftarrow{X}_t)_{t\in[0,1]}$ is Lipschitz with a dimension-free constant, more precisely,*

$$\text{Lip}(\overleftarrow{X}_1(x)) \le \|\nabla \overleftarrow{X}_1(x)\|_{op} \le \exp\left(\frac{K_1 + K_2}{2}\right). \tag{33}$$

*Proof.* Recall the Föllmer flow equation 7 with $\|\nabla V(t,\cdot)\|_\infty \le (K_1 + K_2)t$ in Theorem 8, by following the Theorem 1 in (Mikulincer & Shenfeld, 2023), we arrive at the following result,

$$\text{Lip}(\overleftarrow{X}_1(x)) \le \|\nabla \overleftarrow{X}_1(x)\|_{op} \le \exp\left(\int_0^1 (K_1 + K_2)s ds\right) \le \exp\left(\frac{K_1 + K_2}{2}\right).$$

$\square$

Bound like equation 33 can also be achieved in (Caffarelli, 2000; Colombo et al., 2017; Kim & Milman, 2012; Mikulincer & Shenfeld, 2023; Brigati & Pedrotti, 2024) under various assumptions, as detailed in Appendix A. In general, the constants involved are dimension-free.

We next show that a similar property also holds for the discrete learned flow $(\overleftarrow{Y}_t)_{t\in[0,1]}$ under the regularity of learned velocity field Assumption 10.

**Corollary 24** (Lipschitzness of discrete flow). *The regularity of learned velocity field Assumption 10 implies the Lipschitz property of the learned flow $(\overleftarrow{Y}_t)_{t\in[0,1]}$ with a dimension-free constant, such that*

$$\text{Lip}(\overleftarrow{Y}_1(x)) \le \|\nabla \overleftarrow{Y}_1(x)\|_{op} \le \exp\left(\frac{K_1 + K_2 + K_8}{2}\right).$$

*Proof.* For any $x, y \in \mathbb{R}^d$, $t \in [t_n, t_{n+1}]$ with $k = 0, 1, \dots, N-1$, Itô's formula gives

$$\frac{d|\overleftarrow{Y}_t(x) - \overleftarrow{Y}_t(y)|^2}{dt} = 2\langle \overleftarrow{Y}_t(x) - \overleftarrow{Y}_t(y), \widetilde{V}(t_n, \overleftarrow{Y}_{t_n}(x)) - \widetilde{V}(t_n, \overleftarrow{Y}_{t_n}(y))\rangle.$$

Using the Cauchy-Schwarz inequality, we obtain

$$\frac{d|\overleftarrow{Y}_t(x) - \overleftarrow{Y}_t(y)|^2}{dt} \le 2\sqrt{|\overleftarrow{Y}_t(x) - \overleftarrow{Y}_t(y)|^2}\sqrt{|\widetilde{V}(t_n, \overleftarrow{Y}_{t_n}(x)) - \widetilde{V}(t_n, \overleftarrow{Y}_{t_n}(y))|^2}.$$

Therefore,

$$\frac{d|\overleftarrow{Y}_t(x) - \overleftarrow{Y}_t(y)|}{dt} \le |\widetilde{V}(t_n, \overleftarrow{Y}_{t_n}(x)) - \widetilde{V}(t_n, \overleftarrow{Y}_{t_n}(y))|$$

$$\le \nabla\widetilde{V}|\overleftarrow{Y}_{t_n}(x) - \overleftarrow{Y}_{t_n}(y)|$$

$$\le (K_1 + K_2 + K_8)t_n|\overleftarrow{Y}_{t_n}(x) - \overleftarrow{Y}_{t_n}(y)|,$$

where the last inequality uses Lipschitzness of $\widetilde{V}$ in Assumption 10. Integration over time yields

$$|\overleftarrow{Y}_{t_{n+1}}(x) - \overleftarrow{Y}_{t_{n+1}}(y)|$$
$$\le |\overleftarrow{Y}_{t_n}(x) - \overleftarrow{Y}_{t_n}(y)| + (t_{n+1} - t_n)(K_1 + K_2 + K_8)t_n|\overleftarrow{Y}_{t_n}(x) - \overleftarrow{Y}_{t_n}(y)|.$$

Then, it follows that the Lipschitz constant of the discrete flow satisfies $\text{Lip}(\widetilde{T}_n) \leq 1 + (t_{n+1} - t_n)(K_1 + K_2 + K_8)t_n$. Iterating this bound over all $n = 0, 1, \ldots, N-1$, we obtain the following estimate over the full interval $[0, 1]$,

$$|\overset{\leftarrow}{Y}_1(x) - \overset{\leftarrow}{Y}_1(y)| \leq \left( \prod_{n=0}^{N-1} \text{Lip}(\widetilde{T}_n) \right) |\overset{\leftarrow}{Y}_{t_0}(x) - \overset{\leftarrow}{Y}_{t_0}(y)|$$
$$\leq \exp\left( \frac{K_1 + K_2 + K_8}{2} \right) |x - y|,$$

and conclude the proof. $\qquad\qquad\qquad\qquad\qquad\qquad\qquad\qquad\qquad\qquad\qquad\qquad\square$

### B.8 Proof of Theorem 11

*Proof.* Recall that Assumption 7 ensure the well-podeness and Lipschitzness of Föllmer flow $(\overset{\leftarrow}{X}_t)_{t \in [0,1]}$ in equation 7, with

$$\text{Lip}(T_n) \leq \exp\left( \int_{t_n}^{t_{n+1}} (K_1 + K_2)t \, \mathrm{d}t \right) \quad \text{and} \quad \prod_{j=0}^{N-1} \text{Lip}(T_j) \leq \exp\left( \frac{K_1 + K_2}{2} \right),$$

as established in Lemma 19 and Corollary 23.

Furthermore, Assumption 10 guarantees the Lipschitzness of learned discret Föllmer flow $(\overset{\leftarrow}{Y}_t)_{t \in [0,1]}$ in equation 10, with

$$\text{Lip}(\widetilde{T}_n) \leq 1 + (t_{n+1} - t_n)(K_1 + K_2 + K_8)t_n \text{ and } \prod_{j=0}^{N-1} \text{Lip}(\widetilde{T}_j) \leq \exp\left( \frac{K_1 + K_2 + K_8}{2} \right),$$

as shown in Corollary 24.

Therefore, it only remains to verify that the stepwise approximation error *satisfies Assumption 5*. To analyze the discretization error at each step, we recall the expression in equation 2:

$$T_n(\overset{\leftarrow}{X}_{t_n}) = \overset{\leftarrow}{X}_{t_{n+1}}.$$

Applying the vector-valued Taylor expansion of $\overset{\leftarrow}{X}_{t_{n+1}}$ over $[t_n, t_{n+1}]$, the remainder is defined by

$$R(t) := T_n(\overset{\leftarrow}{X}_{t_n}) - \overset{\leftarrow}{X}_{t_n} - hV(t_n, \overset{\leftarrow}{X}_{t_n}).$$

Under Assumption 7, we can derive the second moment bound by the forward diffusion process equation 5

$$\mathbb{E}_{\bar{p}_t} |\overset{\rightarrow}{X}_t|^2 = \mathbb{E}_{\bar{p}_t} |\overset{\rightarrow}{X}_t - (1-t)\overset{\rightarrow}{X}_0|^2 + \mathbb{E}_{\bar{p}_t} |(1-t)\overset{\rightarrow}{X}_0|^2$$
$$\leq t(2-t)Tr(C) + (1-t)^2 M_2 \leq M_0.$$

Then the expectation of the $R(t)$ is controlled by

$$\mathbb{E}|R(t)|^2 = \mathbb{E}_{\overset{\leftarrow}{X}_{t_n} \sim \overset{\leftarrow}{P}_{t_n}} |T_n(\overset{\leftarrow}{X}_{t_n}) - \overset{\leftarrow}{X}_{t_n} - hV(t_n, \overset{\leftarrow}{X}_{t_n})|^2$$
$$\leq \frac{h^4}{4} \sup_{\tau \in (t_n, t_{n+1})} \mathbb{E}_{\overset{\leftarrow}{X}_\tau \sim \overset{\leftarrow}{P}_\tau} |\partial_\tau V(\tau, \overset{\leftarrow}{X}_\tau)|^2$$
$$\leq \frac{3h^4}{4} \left( K_5^2 M_0 + \frac{K_6^2}{1-t^2} + K_7^2 \right), \quad \forall t \in [0, 1),$$

where the last inequality follows from the bound on $|\partial_t V(t, x)|$ in Theorem 8, which gives

$$\mathbb{E}_{\overset{\leftarrow}{X}_\tau \sim \overset{\leftarrow}{P}_\tau} |\partial_\tau V(\tau, \overset{\leftarrow}{X}_\tau))|^2 \leq 3 \left( K_5^2 M_0 + \frac{K_6^2}{1-\tau^2} + K_7^2 \right).$$

Consequently, the local truncation error is bounded by

$$
\begin{aligned}
&\sqrt{\mathbb{E}_{\overleftarrow{X}_{t_n}\sim\overleftarrow{P}_{t_n}}|T_n(\overleftarrow{X}_{t_n})-\widetilde{T}_n(\overleftarrow{X}_{t_n})|^2}\\
&=\sqrt{\mathbb{E}_{\overleftarrow{X}_{t_n}\sim\overleftarrow{P}_{t_n}}|\overleftarrow{X}_{t_n}+hV(t_n,\overleftarrow{X}_{t_n})+R(t_n)-\overleftarrow{X}_{t_n}-h\widetilde{V}(t_n,\overleftarrow{X}_{t_n})|^2}\\
&\leq\sqrt{h^2\mathbb{E}_{\overleftarrow{X}_{t_n}\sim\overleftarrow{P}_{t_n}}|V(t_n,\overleftarrow{X}_{t_n})-\widetilde{V}(t_n,\overleftarrow{X}_{t_n})|^2}+\sqrt{\mathbb{E}|R(t_n)|^2}\\
&\leq h\left(\frac{\sqrt{3}}{2}\left(K_5\sqrt{M_0}+\frac{K_6}{\sqrt{1-t_n^2}}h+K_7\right)h+\epsilon\right).
\end{aligned}
\tag{34}
$$

The second inequality holds by the error between $\widetilde{V}(t_n,x)$ and $V(t_n,x)$ stated in Assumption 9. This completes the verification of Assumption 5.

Now in Proposition 6, we employ coupling between $\overleftarrow{X}_0 \sim \overrightarrow{P}_1 = \gamma_C$ and $\overleftarrow{Y}_0 \sim \overleftarrow{Q}_0 = \gamma_C$,

$$
\begin{aligned}
\mathcal{W}_2(\overleftarrow{P}_1,\overleftarrow{Q}_1)\leq{}&\exp\left(\frac{K_1+K_2+K_8}{2}\right)\mathcal{W}_2(\overleftarrow{P}_0,\overleftarrow{Q}_0)\\
&+\frac{\exp\left(\frac{K_1+K_2+K_8}{2}\right)-1}{((K_1+K_2+K_8)h)/2}\cdot h\left(\frac{\sqrt{3}}{2}\left(K_5\sqrt{M_0}+K_7\right)h+\epsilon\right)\\
&+\exp\left(\frac{K_1+K_2+K_8}{2}\right)\frac{\sqrt{3}K_6\pi}{4}h\\
\leq{}&\exp\left(\frac{K_1+K_2+K_8}{2}\right)\left(\sqrt{3}\left(K_5\sqrt{M_0}+K_9\right)h+2\epsilon\right).
\end{aligned}
\tag{35}
$$

where a straightforward calculation shows $\sum_{n=0}^{N-1}\frac{K_6}{\sqrt{1-t_n^2}}=\frac{K_6\pi}{2}$; Accordingly, set $K_9:=\frac{K_6\pi}{4}+K_7$. Noting that $\overleftarrow{P}_1 = \overrightarrow{P}_0$, we obtain the conclusion in equation 14. $\qquad\square$

Note that the first term in Proposition 6, stemming from the time-propagating discrepancy of the semi-group maps, vanishes in Theorem 11 because the Föllmer flow $(\overleftarrow{X}_t)_{t\in[0,1]}$ is well-posed at $t=0$, giving $\mathcal{W}_2(\overleftarrow{P}_0,\overleftarrow{Q}_0)=0$. Thus, only the accumulated discretization error remains, corresponding to the second term in Proposition 6.

## B.9 Relation to Prob ODE

The probabilistic ODE (Prob ODE) (Song et al., 2021; Gao & Zhu, 2025)

$$
\frac{d\widehat{X}_s}{ds}=-(\widehat{X}_s-\nabla\log\widehat{p}_s(\widehat{X}_s)),\quad s\in[T,0],
\tag{36}
$$

can be viewed as a time-rescaled Föllmer flow, via $s\mapsto\ln\left(\frac{1}{t}\right)$, where $T$ is finite time. Since Lipschitzness of the transport maps are invariant under time rescaling, the results of Corollary 12 apply directly to the equation 36. The discretization can be chosen as

$$
s_n=\ln\left(\frac{1}{t_n}\right),\quad n=1,\dots,N.
$$

In the forward Prob ODE, the distribution approaches Gaussian only as $s\to+\infty$. To realize this limit in practice, we set $s_0=+\infty$ and initialize the dynamics with $\widehat{X}_{s_0}\sim\mathcal{N}(0,C)$, and take $T=s_1=\ln\left(\frac{1}{t_1}\right)$ with $\widehat{X}_{s_1}=\widehat{X}_{s_0}+t_1\sqrt{C}\mathbb{E}_{\bar{p}_0}[X]$, which corresponds to a single-step first-order Euler method. For $n$-step $(n\geq2)$, the update $\widehat{X}_{s_n}=\widehat{X}_{s_{n-1}}+(e^{s_{n-1}-s_n}-1)\left(S_\theta(e^{-s_{n-1}},\widehat{X}_{s_{n-1}})+\widehat{X}_{s_{n-1}}\right)$ follows the exponential Euler scheme.

## B.10 Proof of convergence theory of 1-rectified flow

*Proof.* The forward process in the first-step rectification (Liu et al., 2023; Rout et al., 2024), constructed by independently coupling the data with a standard Gaussian reference distribution, is defined by the interpolation $\hat{X}_t = \hat{\alpha}_t \hat{X}_1 + \hat{\beta}_t \mathcal{N}$ with $\hat{\alpha}_t = 1 - t, \hat{\beta}_t = t$. Then the transition probability distribution from $\hat{X}_0$ to $\hat{X}_t$ is given by

$$\hat{X}_t | \hat{X}_1 = x_1 \sim \mathcal{N}\big((1-t)x_1, t I_d\big). \tag{37}$$

Under the Gaussian-tail Assumption 7, the score function can be calculated by

$$\hat{S}(t,x) := \nabla \log \hat{p}_t = \nabla \log \int_{\mathbb{R}^d} \big(2\pi \det(\hat{B}(t))\big)^{-\frac{d}{2}} G(t,x,y) \cdot \exp\left(-\frac{|x|_{\hat{A}_t}}{2}\right) \exp\left(h(y)\right) \, \mathrm{d}y,$$

where $G(t,x,y) := \exp\left(-\frac{|\hat{K}(t)x-y|^2_{\hat{B}(t)}}{2}\right)$, $\hat{A}_t = At^2 + (1-t)^2 I_d$, $\hat{K}(t) = (A\hat{A}_t^{-1})t$, $\hat{B}(t) = (A\hat{A}_t^{-1})(1-t)^2$. First, we consider the modified score function:

$$\widetilde{\hat{S}}(t,x) := \hat{S}(t,x) + \hat{A}_t^{-1} x \le |\hat{K}(t)\nabla h(x)|.$$

Let $\hat{K} = \sup_{0 \le t \le 1} |\frac{1}{t}\hat{K}(t)| = \sup_{0 \le t \le 1} |A\hat{A}_t^{-1}| \le 1 + \|A\|$, then we have

$$|\widetilde{\hat{S}}(t,\cdot)|_\infty \le \hat{K}|\nabla h|_\infty t = \hat{K}_0 t$$

with $\hat{K}_0 := \hat{K}|\nabla h|_\infty$.
Taking the derivative twice along that direction and using the same method as above, it yields

$$\|\nabla \widetilde{\hat{S}}(t,\cdot)\|_\infty \le \hat{K}(t)^2(\|\nabla^2 h\|_\infty + |\nabla h|^2_\infty) = \hat{K}_1 t^2,$$

where $\hat{K}_1 := \hat{K}^2(\|\nabla^2 h\|_\infty + |\nabla h|^2_\infty)$. Define $\hat{K}_2 := \sup_{0 \le t \le 1} \|\frac{1}{t}\big(I_d - (1-t)\hat{A}_t^{-1}\big)\| = \sup_{0 \le t \le 1} \|(A + C - I_d)\big(At^2 + (1-t)^2 I_d\big)^{-1}\|$, we obtain

$$|\hat{V}(t,x)| := \left|\frac{x + (1-t)\hat{S}(t,\cdot)}{t}\right| = \left|\frac{(1-t)\widetilde{\hat{S}}(t,\cdot) + \big(I_d - (1-t)\hat{A}_t^{-1}\big)x}{t}\right| \le \hat{K}_0 + \hat{K}_2|x|,$$

and

$$\|\nabla \hat{V}(t,x)\|_\infty = \left\|\nabla\left(\frac{(1-t)\widetilde{\hat{S}}(t,\cdot) + \big(I_d - (1-t)\hat{A}_t^{-1}\big)x}{t}\right)\right\|_\infty \le \hat{K}_1 t + \hat{K}_2.$$

Similar to the proof of $|\partial_t V|$ of Föllmer flow in Appendix B.2, we derive

$$|\partial_t \hat{V}(t,x)| \le \hat{K}_5|x| + \frac{\hat{K}_6}{\sqrt{1-t^2}} + \hat{K}_7.$$

where $\hat{K}_5, \hat{K}_6$ and $\hat{K}_7$ are dimension-free constants. This completes the argument that the trajectory $\hat{X}_t$ of 1-rectified flow possesses the similar regularity properties for its velocity field as those established in Theorem 8. Consequently, by following the proof steps of Theorem 11 in Appendix B.8, the desired result directly follows. □

## B.11 Convergence in the Bayesian Inverse problems

We are aware of several posterior analyses, such as Bayesian inverse problems (van de Schoot et al., 2021), used in uncertainty quantification to infer model parameters $x$ from observations $y \in \mathbb{R}^m$. The posterior typically takes the form of

$$\bar{p}_0(x) = D_0 \exp\left(-\frac{|x|^2_C}{2}\right) \exp\left(-\frac{|G(x)-y|^2_\Sigma}{2}\right), \tag{38}$$

where $D_0$ is a normalizing constant, $C$ denotes the covariance matrix of the Gaussian prior, $\Sigma$ represents the covariance of the observational noise and $G \in C_2^b(\mathbb{R}^d, \mathbb{R}^m)$ is a nonlinear forward operator. In our training framework, we adopt the Gaussian prior with covariance $C$ from equation 38 as the invariant measure of the forward diffusion process equation 5. The conditioned score (Batzolis et al., 2021) in the score matching is trained by minimizing

$$\mathbb{E}_{\bar{p}_t(x;y)}\big|\widetilde{s}(1-t,x;y) - C\nabla_x \log \bar{p}_t(x;y)\big|^2,$$

where $\bar{p}_t$ denotes the joint law of $(X_t, Y)$ with $Y = G(X_0) + \mathcal{N}(0, \Sigma)$. For ODE-based generation of the posterior distribution with observation $y$, we impose the following assumption on the approximation error of the velocity field $V(t, x; y)$ given in equation 7.

**Assumption 25.** *Fixing observation $y$, for each time discretization point $t_n$,*

$$\mathbb{E}_{\overrightarrow{P}_{1-t_n};y}|V(t_n, x; y) - \widetilde{V}(t_n, x; y)|^2 \le \epsilon^2.$$

**Theorem 26.** *Suppose accuracy Assumption 25 and regularity Assumption 10 hold. Using the Euler method to the Föllmer flow with uniform step size $h = t_{n+1} - t_n \le 1$ yields,*

$$\mathcal{W}_2(\overrightarrow{P}_0(\cdot, y), \overleftarrow{Q}_1(\cdot, y)) \le \exp\big(\frac{\widetilde{K_1} + K_8}{2}\big)\left(\frac{\sqrt{3}}{N}\left(\widetilde{K_5}\sqrt{M_0} + \widetilde{K_9}\right) + 2\epsilon\right). \tag{39}$$

*where $\widetilde{K_1}, \widetilde{K_5}, \widetilde{K_9}$ are dimension-free constants depending on $(\|C\|, \|\Sigma\|, \|G\|_{C_2}, |y|)$, see Table 6 of Appendix C , and the constant $K_8$ is defined in Assumption 10.*

*Proof.* Take $A = C$, and $h(x) = -\frac{|G(x)-y|_\Sigma^2}{2}$, then $h(x)$ satisfies,

$$|\sqrt{C}\nabla h(x)| = |\sqrt{C}\nabla G(x)\Sigma^{-1}(G(x)-y)| \le \|C\|^{\frac{1}{2}}\left(|G|_\infty + |y|\right)\|\Sigma^{-1}\|\|\nabla G\|_\infty,$$
$$\|C\nabla^2 h(x)\| = \|C\nabla^2 G(x)\Sigma^{-1}(G(x)-y) + C\nabla G(x)\Sigma^{-1}\nabla G(x)^T\|$$
$$\le \|C\|\|\Sigma\|^{-1}\|\nabla^2 G\|_\infty(|G|_\infty + |y|\|\nabla G\|_\infty^2).$$

Then by Theorem 11, we obtain the bound equation 39 with the constants replaced as specified in Table 6. $\square$

**Remark 27.** *With fixed $\widetilde{K_1}$, $\widetilde{K_5}$ and $\widetilde{K_9}$, for $\epsilon_0$ accuracy in $W_2$ distance for 39, one requires at most:*

$$N = \mathcal{O}\left(\frac{\sqrt{M_0}}{\epsilon_0}\right), \quad \epsilon = \mathcal{O}(\epsilon_0).$$

## C  Summary of notation and dimension-free constants.

In this part, we provide the frequently used notation and the dimension-free constants appearing in Theorem 8, Theorem 11, Theorem 26, Corollary 14 and Theorem 15.

Table 3: Notation for the forward and reverse-time processes.

| Notation | Meaning |
|---|---|
| $\overrightarrow{X}_t$ | Forward diffusion process |
| $\overleftarrow{X}_t$ | backward diffusion process |
| $\bar{p}_t$ | probability distribution function of the forward process |
| $p_t$ | probability distribution function of backward process |
| $\overrightarrow{P}_t$ | marginal distribution of $\overrightarrow{X}_t$ |
| $\overleftarrow{P}_t$ | marginal distribution of $\overleftarrow{X}_t$ |
| $S(t,x)$ | score function |
| $V(t,x)$ | velocity field |
| $\widetilde{V}(t,x)$ | learned velocity field |

Table 4: Explicit for coefficients in Thm. 8 and Thm. 11.

| Coefficient | Explicit expressions |
|---|---|
| $\bar{A}_t$ | $At^2 + C(1-t^2)$ |
| $K(t)$ | $(A\bar{A}_t^{-1})t$ |
| $B(t)$ | $(A\bar{A}_t^{-1})(1-t^2)$ |
| $K$ | $\sup_{0\le t\le 1}|A\bar{A}_t^{-1}| \le \max\{1, \|AC^{-1}\|\}$ |
| $K_0$ | $K\|C\|^{1/2}|\sqrt{C}\nabla h|_\infty$ |
| $K_1$ | $K^2(\|C\nabla^2 h\|_\infty + |\sqrt{C}\nabla h|_\infty^2)$ |
| $K_2$ | $\sup_{0\le t\le 1}\|\frac{1}{t^2}(I - C\bar{A}_t^{-1})\| = \sup_{0\le t\le 1}\|(A-C)(At^2 + C(1-t^2))^{-1}\|$ |
| $K_3$ | $\sup_{0\le t\le 1}\|\frac{A\partial_t(\bar{A}_t^{-1})}{t}\| = \sup_{0\le t\le 1}\|2A(A-C)\big((At^2 + C(1-t^2))^2\big)^{-1}\|$ |
| $K_4$ | $\sup_{0\le t\le 1}\|\frac{C\partial_t(\bar{A}_t^{-1})}{t}\| = \sup_{0\le t\le 1}\|2C(A-C)\big((At^2 + C(1-t^2))^2\big)^{-1}\|$ |
| $K_5$ | $\max\{3K_1C_1, 2(1+\sqrt{2})K_0\sqrt{K_1}C_1\} + K_2 + K_4$ |
| $K_6$ | $2K_1K^{-1}\|C\|^{\frac{1}{2}}C_2$ |
| $K_7$ | $\max\{C_3, C_4\} + K_0K_3K^{-1}$ |
| $K_9$ | $\frac{1}{4}K_6\pi + K_7$ |
| $C_1$ | $\max_{t\in[\sqrt{1-\frac{1}{2\|C\|\|\nabla^2 h\|_\infty}},1)}\{\partial_t K(t)\} = \max\{\frac{\|A\|(\|(A-C)(1-\frac{1}{2\|C\|\|\nabla^2 h\|_\infty})-C\|)}{(\|C\|+\|(A-C)\|(1-\frac{1}{2\|C\|\|\nabla^2 h\|_\infty}))^2}, \frac{\|A-2C\|}{\|A\|}, \frac{\|A\|}{8\|C\|}\}$ |
| $C_2$ | $\max_{t\in[\sqrt{1-\frac{1}{2\|C\|\|\nabla^2 h\|_\infty}},1)}\{\partial_t B(t)\} = \max\{2, \frac{2\|A\|^2\sqrt{1-\frac{1}{2\|C\|\|\nabla^2 h\|_\infty}}}{(\|C\|+\|(A-C)\|(1-\frac{1}{2\|C\|\|\nabla^2 h\|_\infty}))^2}, \frac{9A^2}{8\|C\|^2}\sqrt{\frac{\|C\|}{3\|A-C\|}}\}$ |
| $C_3$ | $2K_1C_2(K_1^{\frac{1}{2}} + K_0)$ |
| $C_4$ | $2\sqrt{3}K_0K_1C_2 + \frac{1}{2}K_0^{\frac{3}{2}}K_1^{\frac{1}{2}}C_2 + \frac{(1+\sqrt{6})^2}{4(1+\sqrt{2})}K_0^2K_1^{\frac{1}{2}}C_2$ |

Table 5: Explicit for coefficients of Gaussian-mixture example in subsection 3.5.

| Coefficient | Explicit expressions |
|---|---|
| $\bar{A}_t$ | $(\tau t^2 + 1 - t^2)I_d$ |
| $K(t)$ | $\frac{\tau t}{\tau t^2 + 1 - t^2}I_d$ |
| $B(t)$ | $\frac{\tau(1-t^2)}{\tau t^2 + 1 - t^2}I_d$ |
| $K$ | $\max\{1, \tau\}$ |
| $K_0$ | $K\frac{|a|_2}{\tau}$ |
| $K_1$ | $2K^2\frac{|a|_2^2}{\tau^2}$ |
| $K_2$ | $\sup_{t\in[0,1]}\frac{|\tau-1|}{\tau t^2 + (1-t^2)}$ |
| $K_3$ | $\sup_{t\in[0,1]}\frac{2\tau|\tau-1|}{(\tau t^2 + 1 - t^2)^2}$ |
| $K_4$ | $\sup_{t\in[0,1]}\frac{2|\tau-1|}{(\tau t^2 + 1 - t^2)^2}$ |
| $K_5$ | $\max\{3K_1C_1, 2(1+\sqrt{2})K_0\sqrt{K_1}C_1\} + K_2 + K_4$ |
| $K_6$ | $2K_1K^{-1}\|C\|^{\frac{1}{2}}C_2$ |
| $K_7$ | $\max\{C_3, C_4\} + K_0K_3K^{-1}$ |
| $K_9$ | $\frac{1}{4}K_6\pi + K_7$ |
| $C_1$ | $\max\left\{\frac{\tau\left|(\tau-1)(1-\frac{\tau^2}{2|a|_2^2})-1\right|}{\left(1+|\tau-1|(1-\frac{\tau^2}{2|a|_2^2})\right)^2}, \frac{|\tau-2|}{\tau}, \frac{\tau}{8}\right\}$ |
| $C_2$ | $\max\left\{2, \frac{2\tau^2\sqrt{1-\frac{\tau^2}{2|a|_2^2}}}{\left(1+|\tau-1|1-\frac{\tau^2}{2|a|_2^2}\right)^2}, \frac{9\tau^2}{8}\sqrt{\frac{1}{3|\tau-1|}}\right\}$ |
| $C_3$ | $2K_1C_2(K_1^{\frac{1}{2}} + K_0)$ |
| $C_4$ | $2\sqrt{3}K_0K_1C_2 + \frac{1}{2}K_0^{\frac{3}{2}}K_1^{\frac{1}{2}}C_2 + \frac{(1+\sqrt{6})^2}{4(1+\sqrt{2})}K_0^2K_1^{\frac{1}{2}}C_2$ |

Table 6: Explicit for coefficients in Thm. 26.

| Coefficient | Explicit expressions |
|---|---|
| $\widetilde{K_0}$ | $\|C\|\|\Sigma^{-1}\|\,\|\nabla G\|_\infty(|G|_\infty + |y|)$ |
| $\widetilde{K_1}$ | $\|C\|\left(\|\Sigma^{-1}\|\left(\|\nabla G\|_\infty^2 + (|G|_\infty + |y|)\|\nabla^2 G\|_\infty\right) + \|\Sigma^{-2}\|\,\|\nabla G\|_\infty^2(\|G\|_\infty + |y|)^2\right)$ |
| $\widetilde{K_5}$ | $\max\{3\widetilde{K_1}, 2(1+\sqrt{2})\widetilde{K_0}\widetilde{K_1}^{\frac{1}{2}}\}$ |
| $\widetilde{K_6}$ | $4\|C\|^{\frac{1}{2}}\widetilde{K_1}$ |
| $\widetilde{K_7}$ | $\max\{\widetilde{C_3}, \widetilde{C_4}\}$ |
| $\widetilde{K_9}$ | $\|C\|^{\frac{1}{2}}\widetilde{K_1}\pi + \widetilde{K_7}$ |
| $\widetilde{C_3}$ | $4\widetilde{K_1}(\widetilde{K_1}^{\frac{1}{2}} + \widetilde{K_0})$ |
| $\widetilde{C_4}$ | $4\sqrt{3}\widetilde{K_0}\widetilde{K_1} + \widetilde{K_0}^{\frac{3}{2}}\widetilde{K_1}^{\frac{1}{2}} + \frac{(1+\sqrt{6})^2}{2(1+\sqrt{2})}\widetilde{K_0}^2\widetilde{K_1}^{\frac{1}{2}}$ |

Table 7: Explicit for coefficients in Cor. 14 and Thm. 15.

| Coefficient | Explicit expressions |
|---|---|
| $\bar{A}_t$ | $(1-t^2)I_d$ |
| $K_0^*$ | $\frac{R}{1-(1-\delta)^2}$ |
| $K_1^*$ | $3\left(\frac{R}{1-(1-\delta)^2}\right)^2$ |
| $K_2^*$ | $\sup_{t\leq 1-\delta}\|\frac{1}{t^2}(I-\bar{A}_t^{-1})\| = \frac{1}{1-(1-\delta)^2}$ |
| $K_3^*$ | $\frac{2}{1-(1-\delta)^2}$ |
| $K_4^*$ | $\frac{2}{(1-(1-\delta)^2)^2}$ |
| $K_5^*$ | $\frac{9C_1^*R^2}{(1-(1-\delta)^2)^2} + \frac{2}{(1-(1-\delta)^2)^2} + \frac{1}{1-(1-\delta)^2}$ |
| $K_6^*$ | $\frac{6C_2^*R^2}{(1-(1-\delta)^2)^2}$ |
| $K_7^*$ | $6(\sqrt{3}+1)\left(\frac{R}{1-(1-\delta)^2}\right)^3 C_2^* + \frac{2R}{(1-(1-\delta)^2)^2}$ |
| $K_9^*$ | $\frac{3\pi}{2}\frac{C_2^*R^2}{(1-(1-\delta)^2)^2} + 6(\sqrt{3}+1)\frac{C_2^*R^3}{(1-(1-\delta)^2)^3} + \frac{2R}{(1-(1-\delta)^2)^2}$. |
| $C_1^*$ | $\max\{\frac{(1-(1-\delta)^2)\|(1-\delta)^2(1-\frac{1}{2\|\nabla^2 h\|_\infty})-1\|}{\|1+(1-\delta)^2(1-\frac{1}{2\|\nabla^2 h\|_\infty})\|^2}, \ \frac{\|-(1-\delta)^2-1\|}{\|1-(1-\delta)^2\|}, \ \frac{\|1-(1-\delta)^2\|}{8}\},$ |
| $C_2^*$ | $\max\{2, \frac{2(1-(1-\delta)^2)^2(1-\frac{1}{2\|\nabla^2 h\|_\infty})^{\frac{1}{2}}}{\|1+(1-\delta)^2(1-\frac{1}{2\|\nabla^2 h\|_\infty})\|^2}, \ \frac{9(1-(1-\delta)^2)^2}{8\sqrt{3}(1-\delta)},$ |

