# OpenReview forum: "Pathway to $O(\sqrt{d})$ Complexity bound under Wasserstein metric of flow-based models"
_TMLR — Under review for TMLR_

### Review · Reviewer_yw9j · 2026-05-19

**Summary Of Contributions:**

The paper establishes an $O(\sqrt{d})$ iteration complexity bound for the Föllmer flow and 1-rectified flow under the Wasserstein-2 metric, without requiring log-concavity of the target distribution. The core framework decomposes the $W_2$ error into the accumulated Lipschitz propagation of the push-forward maps and a local discretization error. Under a Gaussian tail assumption on the target, the Föllmer velocity field is shown to be globally Lipschitz with dimension-free constants, and its time derivative $\partial_t V$ is bounded by $O(|x| + \frac{1}{\sqrt{1-t^2}})$, which is integrable over $[0,1]$. As a result, the flow can be sampled over the entire interval, avoiding the early stopping required in prior work and the corresponding logarithmic penalties in the complexity bound of Wang & Wang (2024). The achieved scaling is shown to be optimal for Gaussian targets. The result represents an improvement over Wang & Wang (2024) under the same Gaussian tail assumption but is largely incomparable with the bounds of Bruno & Sabanis (2025) and Gentiloni-Silveri & Ocello (2025), as the Gaussian tail assumption and their weak log-concavity assumption do not imply one another.

**Additional Comments:**

Questions to the authors:
- Assumption 8 seems very restrictive. Is there a rationale for why we would expect the assumption to hold in practical settings?
- Assumption 14 requires the learned velocity field to be uniformly Lipschitz. How easy is this to impose this on currently used neural network-based models.

**Audience:**

Yes

**Audience Explanation:**

The paper contributes to the theory on the convergence of flow-based generative models, which is an area of active interest to the TMLR community. The presented theoretical analysis is interesting and novel to the best of my knowledge, though its applicability might be limited.

**Claims And Evidence:**

Yes

**Claims Explanation:**

The presented theory appears to be technically sound (I have not verified the proofs) and the authors support their claims with experiments that demonstrate the $O(\sqrt{d})$ dimensional scaling and first-order convergence in step size numerically. However, Assumption 8 and Assumption 14 seem very strong and I'm doubtful as to how applicable the theory is in practice. Assumption 8 does not only require the tails to be Gaussian, but also that the gradient and Hessian of the remainder must be uniformly bounded. This is the case for the GMM in the presented experiment, but a slight modification of the GMM such that $\vert a \vert$ grows with $d$ would already violate it. More generally, Assumption 8 requires the log-density to vary smoothly everywhere, likely excluding targets with rapid density transitions or complex multimodal structure in the bulk of the distribution. Moreover, Assumption 14 requires the learned velocity field to be uniformly Lipschitz, and whether this holds for neural network approximations in practice is unclear.

**Requested Changes:**

I think the paper would benefit from a more detailed discussion on the practical implications of Assumptions 8 and 14.

---

> ### Author Response · Authors · 2026-06-15
>
> We thank the reviewer for the thoughtful comments and valuable suggestions. Below, we address the points raised and indicate the corresponding revisions made in the manuscript.
>
> **Q1** :Assumption 8 seems very restrictive...
>
> We sincerely thank the reviewer for this insightful comment. We agree that Assumption 8 (Assumption 7 in the revised manuscript) imposes stronger regularity than weak semi-log-concavity assumptions commonly used in the literature. The motivation for this assumption is twofold.
> -  Although restrictive, the Gaussian-tail structure is not artificial and remains compatible with practically relevant settings. In particular, it covers non-log-concave distributions arising from standard early-stopping procedures on compactly supported data, as well as infinite-dimensional Bayesian inverse problems (see Theorem 15 and Theorem 26). Similar Gaussian-tail-type assumptions have also appeared in recent work on sampling and score-based models [1].
> - The assumption is introduced to obtain quantitative regularity of the Fölmer velocity field and derive dimension-dependent convergence guarantees. It implies finite moments of all orders, provides explicit control of the Lipschitz constant and time derivative of the velocity field, and enables the change-of-variables argument leading to the $\mathcal{O}(\sqrt d)$ complexity bound. By contrast, weaker assumptions such as semi-log-concavity typically only yield $\mathcal{O}(d)$ rates. We also view relaxing this assumption as an important direction for future work.
>
> **Q2** :Assumption 14 requires the learned velocity field to be uniformly Lipschitz...
>
> We thank the reviewer for the valuable feedback. We agree that uniform Lipschitz regularity of the learned velocity field is a relatively strong assumption and may not hold automatically for unconstrained neural-network parameterizations.
> - From a practical perspective, Lipschitz control is compatible with standard neural network architectures and training techniques. In particular, spectral normalization [2], Lipschitz-constrained parameterizations [3], and controlled residual architectures [4] are widely used to promote smoothness and stabilize score or flow estimation. Therefore, Assumption 14 (Assumption 10 in the revised manuscript) should be viewed as an idealized regularity condition rather than a strict implementation requirement.
>
> - From a theoretical perspective, similar regularity assumptions are standard in convergence analyses of score-based generative models. For example, [5,6] assume global Lipschitz continuity of the learned score together with temporal regularity, while [7] further imposes higher-order smoothness to control discretization errors of probability flow ODEs. Assumption 10 in the revised manuscript serves an analogous role in our setting, ensuring stability from the continuous flow to its discrete approximation. Moreover, the temporal part of this assumption can be relaxed to require only bounded accumulated discrete-time score gradients; see Remark 21 in Appendix B.6. Alternative approaches such as [8] instead impose an $L^2$ score-approximation condition along the sampling trajectory (Assumption H2), which is closer to the sampling process but leads to different analytical trade-offs. In contrast, our formulation provides a more direct stability condition that enables the optimal $\mathcal{O}(\sqrt d)$ dimension dependence (as opposed to the $\mathcal{O}(d)$ dependence in [8]). Whether such uniform Lipschitz control is strictly necessary remains open, but it is essential for closing our current Wasserstein recursion analysis. We have provided a detailed discussion of this point in Appendix B.6 of the revised manuscript.

---

> > ### Author Response · Authors · 2026-06-15
> >
> > **Comment under “Evidence”** : Assumption 8 does not only require the tails to be Gaussian, but also that the gradient and Hessian of the remainder must be uniformly bounded...
> >
> > We thank the reviewer for this insightful observation. We agree that Assumption 8 (Assumption 7 in the revised manuscript) excludes target distributions with sharp density transitions or complex multimodal structures. Our objective is not to characterize the most general class of distributions or provide end-to-end guarantees for arbitrary trained neural flow models, but rather to identify verifiable regularity conditions under which the optimal $\mathcal{O}(\sqrt d)$ Wasserstein sampling complexity can be rigorously established.
> >
> > The Gaussian-tail assumption provides one tractable setting under which dimension-free regularity estimates can be established. At the same time, our analysis is not restricted to this framework. Theorem 15 shows that under early stopping it suffices to assume a bounded-support distribution with radius $R$, which still yields dimension-free estimates when $R$ is independent of the dimension. We note that a related bounded-support setting has also been considered in the recent preprint of Eliot and Francis [9]. This significantly enlarges the class of admissible targets. We also note that if $R$ scales with the dimension, the regularity constants become dimension-dependent, which may cover more complex distributions but falls outside our current analysis. Extending the theory to such settings is an interesting direction for future work.
> >
> > [1] Frank Cole and Yulong Lu. Score-based generative models break the curse of dimensionality in learning a family of sub-gaussian probability distributions. In 12th International Conference on Learning Representations, ICLR 2024, 2024.
> >
> > [2] Takeru Miyato, Toshiki Kataoka, Masanori Koyama, and Yuichi Yoshida. Spectral normalization for generative adversarial networks. arXiv preprint arXiv:1802.05957, 2018.
> >
> > [3] Henry Gouk, Eibe Frank, Bernhard Pfahringer, and Michael Cree. Regularisation of neural networks by enforcing lipschitz continuity. Machine Learning, 110:393–416, 2021
> >
> > [4] Behrmann, Jens, et al. Invertible residual networks. International conference on machine learning. PMLR, 2019.
> >
> > [5] Stefano Bruno and Sotirios Sabanis. Wasserstein convergence of score-based generative models under semi-convexity and discontinuous gradients. Transactions on Machine Learning Research, 2025.
> >
> > [6] Stefano Bruno, Ying Zhang, Dong-Young Lim, Ömer Deniz Akyildiz, and Sotirios Sabanis. On diffusion-based generative models and their error bounds: The log-concave case with full convergence estimates. arXiv preprint arXiv:2311.13584, 2023.
> >
> > [7] Daniel Zhengyu Huang, Jiaoyang Huang, and Zhengjiang Lin. Convergence analysis of probability flow ode for score-based generative models. IEEE Transactions on Information Theory, 2025.
> >
> > [8] Marta Gentiloni-Silveri and Antonio Ocello. Beyond log-concavity and score regularity: Improved convergence bounds for score-based generative models in w2-distance. arXiv preprint arXiv:2501.02298, 2025.
> >
> > [9] Eliot Beyler and Francis Bach. Convergence of deterministic and stochastic diffusion-model samplers: Asimple analysis in wasserstein distance. arXiv preprint arXiv:2508.03210, 2025.

---

### Review · Reviewer_MnZ8 · 2026-05-24

**Summary Of Contributions:**

This paper studies deterministic flow-based generative models, in which a simple base distribution is transported to a target distribution through a sequence of deterministic maps. Specifically, the authors consider the initial value problem
$$
\frac{d}{dt}x_t = V(t,x_t), \qquad t\in[0,1], \quad x_t\in\mathbb{R}^d,
$$
where the solution flow induced by the time-dependent vector field $V$ defines a transport map from the initial distribution to the target distribution. The continuous-time flow is discretised on a time grid
$
0=t_0<t_1<\cdots<t_{N-1}< t_N =1,
$
so that the overall transport is represented as a composition of local flow maps. Denoting by $T_i:\mathbb{R}^d\to\mathbb{R}^d\$ the exact flow map over the interval $[t_i,t_{i+1}]$, the implemented sampler replaces each $T_i$ with an approximate map $\widetilde{T}_i$. The paper analyses how the local approximation errors of these maps propagate through the composition and affect the final $2$-Wasserstein error between the generated and target distributions.

The main result shows that, under suitable regularity and approximation assumptions, the final $2$-Wasserstein error of the discretized sampler admits an upper bound whose dimension dependence is of order $\sqrt d$ when the base distribution is an isotropic Gaussian. More generally, the bound depends on the covariance of the base distribution through a trace-type quantity. The authors then apply this general error-propagation result to Föllmer flow models, deriving corresponding Wasserstein error bounds for samplers based on approximations of the Föllmer velocity field.

**Additional Comments:**

The dimension-free control of constants such as $K_2$ seems to rely on strong assumptions about the relationship between the Gaussian component of the target distribution and the Gaussian base distribution. In particular, the matrices $ A $ and $ C $, which encode the Gaussian tail structure of the target and the covariance of the base law, are assumed to be simultaneously diagonalisable and uniformly comparable. This means that the base distribution must be well aligned with the Gaussian geometry of the target. The paper should make clearer that the resulting dimension-free Lipschitz bounds are not intrinsic to arbitrary targets, but depend on choosing a base Gaussian whose covariance structure is compatible with the target’s Gaussian tail matrix (if this is the case).

**Audience:**

Yes

**Audience Explanation:**

Flow-based generative models are popular in ML and therefore this paper lies in the scope of TMLR.

**Claims And Evidence:**

Yes

**Claims Explanation:**

The derived $\sqrt{d}$-dependence seems to be correct

**Requested Changes:**

The major weakness of the paper is its readability.
Overall the paper is currently dense, and the presentation of the results should be improved.
The general result (Theorem 6) does not seem so surprising; the paper could focus on the main result for Föllmer flow,
and perhaps Theorem 16 could be stated much earlier. I find it difficult to find what the main claim of the paper, and the numerous assumptions and corollaries add complexity.
Some concrete results can be separately written into a section dedicated to technical results (like a proof section).

See specific comments:

1. The transition between the generic ODE formulation and the Föllmer flow discussion seems abrupt. The authors should clarify why they introduce Föllmer flows here, e.g., explain the role of this class of models in the paper.
2. The general notation paragraph should be introduced earlier. Some symbols are used even with definitions; for example $\gamma_C$ seems undefined at its first appearance. The symbol $ \mathcal{N} $ in $ X_t $ is also undefined.
3. Some differential "d" are Roman and others are Italian. Please make them consistent (I would suggest Roman)
4. Assumption 2: "We further denote..." can be moved out of this assumption, since this is not an assumption. Also weird English.
5. Assumptions 3 and 4: I would not start a sentence with a math symbol.
6. Assumption 7 seems redundant since all the presented results assume stronger Assumption 8. You could simply mention that a third moment suffices, but do not need to make it a formal assumption.

The authors claim $ O(\sqrt{d}) $ bounds, but the bound depends on some exponential dependence on some constants.
This makes me wonder how tight these bounds are. It would be useful to explicitly compute these constants in a specific example.
This also relates to Numerical Illustration. It is unclear what the coupled estimates is intended to do: what are even $ X^{(i)} $ and $ Y^{(i)} $?
If this aims to serve as a bound $W_2$,  what do we gain from comparing an (possibly loose) upper bound against the derived bound?

---

> ### Author Response · Authors · 2026-06-15
>
> We thank the reviewer for the careful evaluation and helpful suggestions. We address the main concerns below and describe the revisions incorporated into the manuscript.
>
> **Requested Changes 1**：The major weakness of the paper is its readability...
>
> We thank the reviewer for the careful reading and constructive feedback regarding readability and presentation. We fully agree that the original structure did not sufficiently highlight the main contributions, and that the large number of intermediate assumptions and corollaries may have obscured the main message.
>
> We clarify that Theorem 6 (Proposition 6 in the revised manuscript) provides a general error-propagation framework for Wasserstein dynamics under Lipschitz transport maps, while our main contribution is its specialization to the Fölmer flow under the Gaussian-tail assumption, together with dimension-explicit regularity estimates leading to the optimal $\mathcal{O}(\sqrt d)$ complexity bound. A key technical ingredient is the temporal regularity estimate
> $$
> |\partial_t V(t,x)| \lesssim |x| + \frac{1}{\sqrt{1-t^2}} + 1,
> $$
> which controls accumulation errors and avoids early-stopping arguments.
> We also include an informal statement of the main result in the Introduction to improve readability and avoid requiring the reader to go through the full technical development before understanding the main claim.
>
> In the revised manuscript, we have reorganized Section 3 to improve clarity: Theorem 8 is now presented as the first main result for the continuous analysis,  followed by Assumptions 9-10 for discretization and approximation error control, leading to convergence result in Theorem 11. Several intermediate results have been moved to the appendix B.5-B.7 to streamline the presentation.
>
> **Requested Changes 2**：Specific comments.
>
> We thank the reviewer for the comment and have corrected the sentence as suggested.
> - We agree that the transition from the generic ODE framework to the Fölmer flow was too abrupt in the original version. To improve clarity, we added an explicit transition sentence (page 4) clarifying that the generic ODE framework is introduced to establish convergence estimates for general continuous-flow matching models, while the subsequent sections specialize it to the Fölmer flow as the main concrete example in this paper.
> - In the revised manuscript, we clarified these notations directly in the text by explicitly stating that $\mathcal N(0,C)$ denotes the centered Gaussian distribution with covariance matrix $C$, and that $\gamma_C := \mathcal N(0,C)$ denotes the corresponding Gaussian initialization distribution.
> - We have standardized the notation for differential operators throughout the paper and consistently use Roman font (e.g., $\mathrm{d}t$, $\mathrm{d}W_t$) for all differentials in the revised manuscript.
> - In the revised manuscript, we rewrote Assumption using standard sentence structure and removed the phrase “We further denote”. We also revised Assumptions 3 and 4 in original manuscript to avoid starting with mathematical symbols and merged them into a single assumption with parts (i) and (ii) (now Assumption 4 in the revised manuscript), improving readability and consistency throughout the paper.
> - Indeed, the reviewer’s comment helped us recognize that the Gaussian-tail Assumption 7 in the revised manuscript implies finite moments of all orders and, in particular, the second- and third-moment conditions used in the analysis. These conditions therefore need not be imposed separately. We have added a detailed proof in Appendix B.2 in the revised version.

---

> > ### Author Response · Authors · 2026-06-15
> >
> > **Requested Changes 3**: The authors claim $O(\sqrt{d})$ bounds, but the bound depends on some exponential dependence on some constants...
> >
> > We thank the reviewer for this insightful comment.
> >   We agree that the constants in our bound may depend strongly, even exponentially, on certain problem-dependent regularity parameters. However, this phenomenon is not merely an artifact of our proof, but is typical in Lipschitz change-of-variable or transport-map estimates. For example, Remark 1.5 of~[1] considers the one-dimensional target density
> > $$
> > \mu(x)=\exp\bigl(-V(x)-H(x)\bigr),
> > \quad
> > V(x)=\frac{x^2}{2},
> > \quad
> > H(x)=L|x|+\log Z,
> > $$
> > where $Z$ is the normalizing constant.
> > The reference distribution is the standard Gaussian measure $\gamma$ on $\mathbb{R}$. It was shown that any transport map $T$ satisfying $T_{\sharp}\gamma = \mu$ must have Lipschitz norm at least $\exp(L^2/2)$. This example demonstrates that large Lipschitz constants may be intrinsic even in simple one-dimensional settings. Therefore, our $\mathcal{O}(\sqrt d)$ result should be understood as characterizing the dimension dependence after fixing the regularity constants, rather than claiming sharpness of all multiplicative constants.  In the revised manuscript, we have added explicit constant computations for the Gaussian-mixture example in Section 3.5 (Appendix C, Table 5).
> >
> >  Regarding the coupled estimator, $X^{(i)}$ and $Y^{(i)}$ denote samples generated by the continuous dynamics and its Euler discretization under shared initialization. This pathwise coupling yields a computable upper bound on the Wasserstein distance in Theorem 11 and is standard for analyzing discretization error. While other couplings could in principle yield tighter Wasserstein bounds, they are generally inaccessible in practice, especially when the learned velocity field is represented by a neural network and the optimal coupling is unknown. Our goal is therefore not to estimate $\mathcal{W}_2$ sharply, but to verify the predicted dimension dependence in practice. In particular, the observed $\sqrt d$ scaling is consistent with the optimal dimension dependence identified in Proposition~6 of Gao et al. [2] for the Gaussian target case, which corresponds to the special setting of our framework with vanishing perturbation $h=0$ and matched covariance matrices $A=C$.
> >
> > **Additional Comments**: The dimension-free control of constants such as $K_2$ seems to rely on strong assumptions...
> >
> > We thank the reviewer for this important observation. We agree that the dimension-free control of constants such as $K_2$ relies on structural assumptions in the Gaussian-tail framework, and does not hold for arbitrary pairs of covariance matrices $(A,C)$. In particular, Assumption 7 in the revised manuscript requires the Gaussian-tail matrix $A$ and the reference covariance matrix $C$ to be uniformly comparable (and simultaneously diagonalizable in our analysis), ensuring that the relevant spectral quantities remain dimension-independent. We have clarified this point immediately after Assumption 7 in the revised manuscript, explicitly stating that the dimension-free constants arise from this compatibility condition rather than being intrinsic to general target distributions.
> >
> > Such compatibility assumptions are also standard in high-dimensional MCMC and Bayesian inverse problems, where dimension-robust algorithms typically require the proposal or reference Gaussian measure to be aligned with the target geometry; see, for example, the preconditioned Crank-Nicolson method in [3].
> >
> > [1] Giovanni Brigati and Francesco Pedrotti. Heat flow, log-concavity, and lipschitz transport maps. arXivpreprint arXiv:2404.15205, 2024.
> >
> > [2] Xuefeng Gao, Hoang M Nguyen, and Lingjiong Zhu. Wasserstein convergence guarantees for a general class of score-based generative models. Journal of machine learning research, 26(43):1–54, 2025.
> >
> > [3] Simon L Cotter, Gareth O Roberts, Andrew M Stuart, and David White. Mcmc methods for functions:modifying old algorithms to make them faster. Statistical Science, pp. 424–446, 2013.

---

### Review · Reviewer_BXm5 · 2026-06-03

**Summary Of Contributions:**

The paper’s main contribution is a Wasserstein error analysis that gives an $O(\sqrt d)$ sampling complexity bound for certain flow-based generative models. The authors contribute a transport-map-based framework showing how local discretization errors accumulate through Lipschitz flow maps. They then show that, for Föllmer flow under a Gaussian-tail assumption, the required Lipschitz and temporal regularity bounds can be controlled in a dimension-free way. This leads to the claimed $O(\sqrt d/\epsilon_0)$ step complexity under the standard Gaussian prior. The paper also discusses extensions to 1-rectified flow, bounded-support targets, and Bayesian inverse problems.

A strength of the work is that it targets an important theoretical gap: Wasserstein convergence guarantees with favorable dimension dependence beyond standard log-concave settings. The analysis also highlights the role of temporal velocity regularity, which is useful for understanding discretization error in flow-based samplers.

However, the result relies on fairly strong assumptions, especially the Gaussian-tail condition and the assumed accuracy and Lipschitz regularity of the learned velocity field. This creates a gap between the paper’s broad framing around practical flow-based generative models and what the main theorem actually establishes. In my view, the main result is better understood as a conditional sampling/discretization complexity guarantee, rather than an end-to-end guarantee for trained neural flow models.

**Audience:**

Yes

**Audience Explanation:**

The claimed $O(\sqrt d)$ bound and the focus on velocity regularity are theoretically relevant to diffusion/flow-based models, Wasserstein convergence, and sampling complexity. However, the practical impact is limited by strong assumptions and narrow experiments.

**Claims And Evidence:**

No

**Claims Explanation:**

The paper supports a narrower claim than the way it is framed. The main theoretical result appears to be a conditional sampling/discretization complexity guarantee: if the target satisfies the Gaussian-tail assumption, and if the learned velocity field is already accurate and Lipschitz regular, then the sampler achieves the claimed $O(\sqrt d)$ Wasserstein complexity.

However, these assumptions are quite strong, and the paper does not fully justify how realistic they are for practical models. The experimental evidence is a toy example with an explicit velocity field, so it mainly validates the discretization and dimension-scaling behavior under idealized conditions, rather than the broader practical setting suggested by the paper.

**Requested Changes:**

Please feel free to push back.

Critical:
- Clarify the scope of the main result. The paper should state more clearly that the $O(\sqrt d)$ bound depends on assumed velocity accuracy and Lipschitz regularity, rather than being an end-to-end guarantee for trained neural flow models.
- Better justify the Gaussian-tail assumption. Assumption 8 is essential to the result, but its scope is unclear. The paper should show what useful non-log-concave distributions it covers, beyond the favorable Gaussian-mixture example. It should also discuss examples where the assumption fails and whether the hidden constants can grow with dimension.
- Clarify the proof of the key velocity regularity result, especially the bound on $|\partial_t V(t,x)|$. This bound drives the discretization error and the final $O(\sqrt d)$ complexity result. The current proof is hard to audit because it relies on long algebraic estimates, many constants, and several compressed steps where dimension dependence could potentially be hidden. The authors should reorganize this part into clearer intermediate lemmas and explicitly state where each dimension-free constant comes from.

Good to have:
- Add more discussion of the learned velocity field assumption, especially whether it can be expected from common neural network training procedures.
- Clarify the role and limitations of the numerical experiments. The current toy example supports the discretization and dimension-scaling claims, but not the broader practical setting.
- Polish notation and writing. Section 2 uses many similar objects for forward/backward time, such as $\overrightarrow X_t$, $\overleftarrow X_t$, $p_t$, $p_{1-t}$, $\overrightarrow P_t$, and $\overleftarrow P_t$, without a notation table. The proofs also introduce many constants $K_i$ and $C_i$. They make the main argument hard to track.

---

> ### Author Response · Authors · 2026-06-15
>
> We are grateful to the reviewer for the detailed assessment and insightful comments. Below, we clarify the issues raised by the reviewer and explain how they have been addressed in the revised manuscript.
>
> **Critical 1**: Clarify the scope of the main result...
>
> We thank the reviewer for this careful and constructive comment. We agree that the claim should be framed more narrowly. The main result (Theorem 11 in the revised manuscript) provides a conditional sampling complexity guarantee: under the Gaussian-tail assumption on the target distribution and assuming that the learned velocity field is sufficiently accurate and Lipschitz regular, the proposed sampler achieves the stated $\mathcal{O}(\sqrt d)$ Wasserstein complexity. More generally, our work does not aim to provide end-to-end guarantees for neural-network training procedures. Rather, it identifies verifiable regularity and approximation conditions under which the optimal ($\mathcal{O}(\sqrt d)$) dimension dependence can be established. Following the reviewer’s suggestion, we have revised the Abstract and the discussion of Theorem 11 to make this scope explicit.
>
> We further clarify the role of the Gaussian-tail assumption 7 in the revised manuscript, which provides a tractable setting for establishing regularity of the score and velocity fields and enabling the $\mathcal{O}(\sqrt d)$ bound. It covers settings such as early stopping for compactly supported distributions and infinite-dimensional Bayesian inverse problems (Theorem 15 and Theorem 26), and should be viewed as a verifiable regularity condition. It is also compatible with standard Lipschitz-promoting techniques such as spectral normalization [1] and controlled residual architectures.
>
> **Critical 2**: Better justify the Gaussian-tail assumption...
>
> We appreciate this helpful suggestion. We agree that the scope of the Gaussian-tail Assumption 7 in the revised manuscript should be made more explicit. Beyond the Gaussian-mixture example, this assumption covers non-log-concave settings arising from early stopping applied to compactly supported distributions (Theorem 15) and infinite-dimensional Bayesian inverse problems (Theorem 26). We have also clarified that the dimension-free constants are not intrinsic to arbitrary target distributions. They rely on the compatibility between the Gaussian-tail matrix $A$ and the covariance matrix $C$ of the Gaussian reference measure. When this compatibility deteriorates, the regularity constants may grow with dimension and the resulting bound becomes less favorable. Similar compatibility requirements are common in dimension-robust MCMC and Bayesian inverse-problem methods, such as preconditioned Crank-Nicolson [2]. We discuss this point explicitly in the revised manuscript and include Table 5 in Appendix C for the numerical experiments in Section 3.5, illustrating the dependence of the error on the Gaussian-tail regularity constants in the Gaussian-mixture example.
>
> **Critical 3**: Clarify the proof of the key velocity regularity result...
>
> We thank the reviewer for this helpful comment. We agree that the proof of the velocity regularity result, especially the estimate of $|\partial_t V|$, should be easier to audit. In the revised manuscript, we have reorganized the proof of Theorem 8 and made the dependence of all constants explicit. In particular, we now state that the dimension-free constants arise only from: (i) operator-norm bounds involving the Gaussian-tail matrix $A$ and the reference covariance $C$, (ii) the regularity quantities $| \sqrt{C}\nabla h|_\infty$ and $\lVert C\nabla^2 h\\lVert _\infty$ in Assumption 7,  and (iii) the Brascamp-Lieb inequality used to control $|\partial_t V|$. The explicit expressions of these constants are collected in Appendix C, and their roles are summarized around Theorem 8 and at the beginning of its proof.  Moreover, Proposition 6 in revised anuscript shows that the accumulation term is itself dimension-independent. This reflects a general property of Lipschitz change-of-variables and contraction-based estimates, where error propagation is controlled by dimension-free Lipschitz constants. The only explicit dimension dependence comes from the local truncation term, which scales as $\sqrt{\operatorname{Tr}(C)}$. Hence, when $C=I_d$, the final complexity bound scales as $\mathcal{O}(\sqrt d)$ with no hidden dependence on $d$ in the accumulation constants. We state this point below Corollary 12.

---

> > ### Author Response · Authors · 2026-06-15
> >
> > **Good to have 1**: Add more discussion of the learned velocity field assumption...
> >
> > We thank the reviewer for the valuable feedback. We agree that uniform Lipschitz regularity of the learned velocity field is a relatively strong assumption and may not hold automatically for unconstrained neural-network parameterizations.
> > - From a practical perspective, Lipschitz control is compatible with standard neural-network architectures and training techniques. Methods such as spectral normalization [1], Lipschitz-constrained parameterization [3], and controlled residual architectures [4] are commonly used to promote smoothness and stabilize score or flow estimation. Thus, Assumption 14 (Assumption 10 in the revised manuscript) should be viewed as an idealized regularity condition rather than a strict implementation requirement.
> > - From a theoretical perspective, similar regularity assumptions are standard in convergence analyses of score-based generative models. For example, [5,6] assume global Lipschitz continuity of the learned score function with respect to the input variable together with temporal regularity (Assumption 3.a), while [7] further require bounded first-order and second-order derivatives of the learned score function (Assumption 3.3) to control discretization errors of probability flow ODEs. Assumption 10 in the revised manuscript serves an analogous purpose in our setting and is introduced to transfer stability from the continuous flow to its discrete approximation.
> > Moreover, Assumption 10 can be relaxed in the temporal direction by requiring only that the accumulated discrete-time sum of gradient norms remains bounded. Alternatively, one may replace this stability condition with trajectory-level approximation assumptions, such as the $L^2$ score-approximation condition H2 in [8]. As discussed in [8], H2 can be related to standard score-matching estimation errors under an additional spatial Lipschitz assumption on the learned score estimator. The two approaches involve different trade-offs: H2 is closer to the sampling process, whereas our forward-process formulation together with Assumption 10 provides a tractable stability condition for closing the Wasserstein recursion and obtaining the optimal $\mathcal{O}(\sqrt d)$ dimension dependence, improving upon the $\mathcal{O}(d)$ dependence in [8]. Whether such Lipschitz control is intrinsically necessary remains open; see Remark 21 and Remark 22 in Appendix B.6 for details.
> >
> > **Good to have 2**: Clarify the role and limitations of the numerical experiments...
> >
> > We sincerely thank the reviewer for this valuable comment. We agree that the numerical experiments should be interpreted within their intended scope. They are designed to isolate and validate the theoretical predictions on discretization error and dimension scaling, rather than to assess the end-to-end performance of trained neural flow models. We use controlled experiments because neural-network-based settings introduce additional factors, such as architecture, width, optimization error, and statistical estimation error, which may vary with dimension and obscure the discretization effect. Thus, the experiments directly test the predicted first-order discretization behavior and the $\mathcal{O}(\sqrt d)$ dimension scaling.
> >
> > **Good to have 3**: Polish notation and writing...
> >
> > We thank the reviewer for this helpful suggestion. To improve readability, we have added a summary table of frequently used notation in Section~2 (Table 3) and clarified the role of the existing dimension-free constant table in Appendix C (Table 4). These additions make the notation and the role of the various constants more transparent, thereby improving the overall clarity of the manuscript.
> >
> > [1] Takeru Miyato et al.  Spectral normalization for generative adversarial networks. arXiv preprint arXiv:1802.05957, 2018.
> >
> > [2] Simon L Cotter et al.  Mcmc methods for functions:modifying old algorithms to make them faster. Statistical Science, 2013.
> >
> > [3] Henry Gouk et al.  Regularisation of neural networks by enforcing lipschitz continuity. Machine Learning,  2021.
> >
> > [4] Behrmann Jens  et al. Invertible residual networks. International conference on machine learning. PMLR, 2019.
> >
> > [5] Stefano Bruno et al. Wasserstein convergence of score-based generative models under semi-convexity and discontinuous gradients. Transactions on Machine Learning Research, 2025.
> >
> > [6] Stefano Bruno et al.  On diffusion-based generative models and their error bounds: The log-concave case with full convergence estimates. arXiv preprint arXiv:2311.13584, 2023.
> >
> > [7] Daniel Zhengyu Huang et al.  Convergence analysis of probability flow ode for score-based generative models. IEEE Transactions on Information Theory, 2025.
> >
> > [8] Marta Gentiloni-Silveri et al. Beyond log-concavity and score regularity: Improved convergence bounds for score-based generative models in w2-distance. arXiv preprint arXiv:2501.02298, 2025.

---

### Author Response · Authors · 2026-06-15

We thank the reviewers for their insightful feedback. We have thoroughly revised the manuscript in response to the comments, and we believe that the changes improve the clarity and strengthen the presentation of the work. We hope these revisions address the reviewers’ concerns, and we would be happy to continue the discussion if further clarification is needed.